# NETWORK ALIGNMENT WITH TRANSFERABLE GRAPH AUTOENCODERS

## ABSTRACT

Network alignment is the task of establishing one-to-one correspondences between the nodes of different graphs and finds a plethora of applications in high-impact domains. However, this task is known to be NP-hard in its general form, and existing algorithms do not scale up as the size of the graphs increases. To tackle both challenges we propose a novel generalized graph autoencoder architecture, designed to extract powerful and robust node embeddings, that are tailored to the alignment task. We prove that the generated embeddings are associated with the eigenvalues and eigenvectors of the graphs and can achieve more accurate alignment compared to classical spectral methods. Our proposed framework also leverages transfer learning and data augmentation to achieve efficient network alignment at a large scale without retraining. Extensive experiments on both network and sub-network alignment with real-world graphs provide corroborating evidence supporting the effectiveness and scalability of the proposed approach.

## 1 INTRODUCTION

Network alignment, also known as graph matching, is a classical problem in graph theory, that aims to find node correspondence across different graphs and is vital in a number of high-impact domains (Emmert-Streib et al., 2016). In social networks, for instance, network alignment has been used for user deanonymization (Nilizadeh et al., 2014) and analysis (Ogaard et al., 2013), while in bioinformatics it is a key tool to identify functionalities in protein complexes (Singh et al., 2008), or to identify gene–drug modules (Chen et al., 2018). Graph matching also finds application in computer vision (Conte et al., 2003), sociology (Racz & Sridhar, 2021), or politics (Li et al., 2022), to name a few. Graph matching can be cast as a quadratic assignment problem (QAP), which is in general NP-hard (Koopmans & Beckmann, 1957).

Various approaches have been developed to tackle network alignment and can be divided into two main categories; i) optimization algorithms that attempt to approximate the QAP problem by relaxing the combinatorial constraints, ii) embedding methods that approach the problem by implicitly or explicitly generating powerful node embeddings that facilitate the alignment task. Optimization approaches, as (Anstreicher & Brixius, 2001; Vogelstein et al., 2015) employ quadratic programming relaxations, while (Klau, 2009) and (Peng et al., 2010) utilize semidefinite or Lagrangian-based relaxations respectively. Successive convex approximations were also proposed by (Konar & Sidiropoulos, 2020) to handle the QAP. Challenges associated with these methods include high computational cost, infeasible solutions, or nearly optimal initialization requirements. Embedding methods, on the other hand, overcome these challenges, but they usually produce inferior solutions, due to an inherent trade-off between embedding permutation-equivariance and the ability to capture the structural information of the graph. Typical embedding techniques include spectral and factorization methods (Umeyama, 1988; Feizi et al., 2019; Zhang & Tong, 2016; Kanatsoulis & Sidiropoulos, 2022), structural feature engineering methods (Berlingerio et al., 2013; Heimann et al., 2018), and random walk approaches (Perozzi et al., 2014; Grover & Leskovec, 2016a). Recently (Chen et al., 2020; Karakasis et al., 2021) have proposed joint node embedding and network alignment, to overcome these challenges, but these methods do not scale up as the size of the graph increases.

Graph Neural Networks (GNNs) are powerful architectures that learn graph representations (embeddings). They have shown state-of-the-art performance in several tasks, including biology (Gainza et al., 2020; Strokach et al., 2020; Jiang et al., 2021), quantum chemistry (Gilmer et al., 2017), social

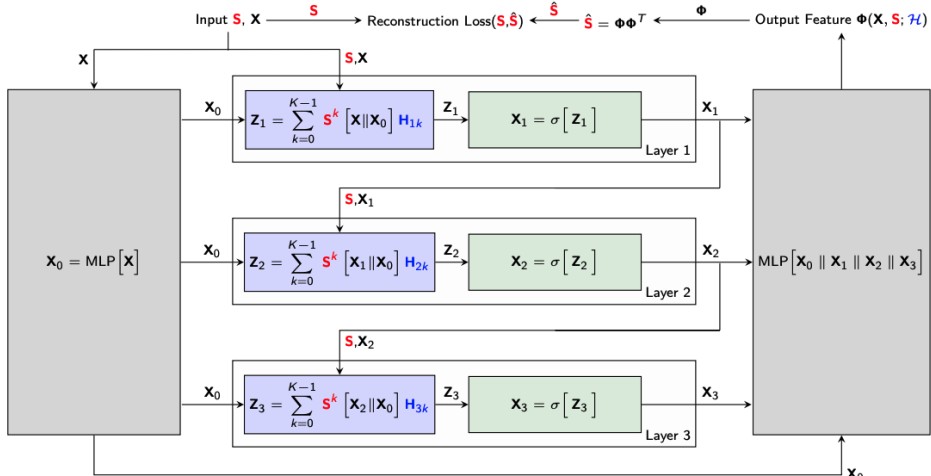

Figure 1: The overall architecture of a three-layer T-GAE and its training paradigm: The input signal is processed by a local MLP and then processed by GNN layers with skip connections. The outputs of all neural network layers are encoded by an MLP followed by a single-layer decoder to generate the reconstructed graph. The whole model is trained end-to-end with a single or multiple graphs.

networks and recommender systems (Ying et al., 2018; Wu et al., 2020). Recently, (Gao et al., 2021a) proposed a GNN approach to match attributed graphs. The method used a joint embedding framework for pairs of graphs and achieved high levels of matching accuracy. However, this method does not scale to large graphs, since training graphs with large sizes is computationally prohibitive.

To address these challenges, we propose a novel self-supervised GNN framework to perform network alignment on a large scale. Specifically, we design a generalized transferable graph autoencoder (T-GAE) (shown in Fig. 1), to produce permutation equivariant and highly expressive embeddings, overcoming the challenges of other embedding techniques. T-GAE is trained on multiple graphs and learns node representations which are tailored to perform alignment between nodes of different graphs. The T-GAE representations combine the eigenvectors of the graph in a nonlinear fashion and are provably at least as good in network alignment as certain spectral methods. Additionally, the proposed framework leverages transfer learning and data augmentation to efficiently operate with large graphs. Training is performed with small graphs, in a self-supervised manner, and the trained encoder can be executed on large graphs to tackle network alignment at a large scale. Extensive experiments with real-world benchmarks test the effectiveness and limits of the proposed T-GAE approach in the tasks of graph and sub-graph matching. The experimental results provide corroborating evidence that T-GAE offers an elegant framework for large-scale network alignment. Our contributions are summarized as follows:

(C1) We propose T-GAE, a generalized graph autoencoder architecture that can be trained with multiple graphs and produce expressive/permutation equivariant representations, tailored to network alignment.

(C2) We draw the the connection between T-GAE and spectral methods and prove that T-GAE is at least as good in graph matching as the absolute value of the graph eigenvectors.

(C3) We leverage data augmentation and transfer learning, to develop a robust framework that efficiently performs network alignment at a large scale.

(C4) We demonstrate the effectiveness and scalability of the proposed T-GAE with real-world, benchmark graphs in challenging graph and sub-graph matching settings.

## 2    PRELIMINARIES

Graphs are represented by $\mathcal{G} := (\mathcal{V}, \mathcal{E})$, where $\mathcal{V} = \{1, \ldots, N\}$ is the set of vertices (nodes) and $\mathcal{E} = \{(v, u)\}$ correspond to edges between pairs of vertices. A graph is represented in a matrix

form by a graph operator $\boldsymbol{S} \in \mathbb{R}^{N \times N}$, where $\boldsymbol{S}(i, j)$ quantifies the relation between node $i$ and node $j$ and $N = |\mathcal{V}|$ is the total number of vertices. In this work, we use the graph adjacency and normalized graph adjacency. Oftentimes, the nodes of the graph are associated with graph signals or node attributes $\boldsymbol{X} \in \mathbb{R}^{N \times D}$, that encode additional information about the nodes. In this paper, we study both network alignment of graphs with or without attributes.

## 2.1 Network Alignment

**Definition 1** (Network Alignment). *Given a pair of graphs $\mathcal{G} := (\mathcal{V}, \mathcal{E})$, $\hat{\mathcal{G}} := (\hat{\mathcal{V}}, \hat{\mathcal{E}})$, with graph adjacencies $\boldsymbol{S}$, $\hat{\boldsymbol{S}}$, network alignment aims to find a bijection $g : \mathcal{V} \to \hat{\mathcal{V}}$ which minimizes the number of edge disagreements between the two graphs. Formally, the problem can be written as:*

$$\min_{\boldsymbol{P} \in \mathcal{P}} \ \left\| \ \boldsymbol{S} - \boldsymbol{P} \hat{\boldsymbol{S}} \boldsymbol{P}^T \ \right\|_F^2 , \tag{1}$$

*where $\mathcal{P}$ is the set of permutation matrices.*

As mentioned in the introduction, network alignment, is equivalent to the QAP, which has been proven to be NP-hard (Koopmans & Beckmann, 1957).

## 2.2 Spectral Decomposition of the Graph

A popular approach to tackle network alignment is by learning powerful node embeddings associated with connectivity information in the graph. Network alignment can be achieved by matching the node embeddings of different graphs rather than graph adjacencies, as follows:

$$\min_{\boldsymbol{P} \in \mathcal{P}} \ \left\| \ \boldsymbol{E} - \boldsymbol{P} \hat{\boldsymbol{E}} \ \right\|_F^2 , \tag{2}$$

where $\boldsymbol{E} \in \mathbb{R}^{N \times F}$ is embedding matrix and $\boldsymbol{E}[i, :]$ is the vector representation of node $i$. The optimization problem in (2) is a linear assignment problem and can be optimally solved in $\mathcal{O}\left(N^3\right)$ by the Hungarian method (Kuhn, 1955b). Simpler sub-optimal alternatives also exist that operate with $\mathcal{O}\left(N^2\right)$ or $\mathcal{O}\left(N \log(N)\right)$ flops.

A question that naturally arises is how to generate powerful node embeddings that capture the network connectivity and also be effective in aligning different graphs. A natural and effective approach is to leverage the spectral decomposition of the graph, $\boldsymbol{S} = \boldsymbol{V} \boldsymbol{\Lambda} \boldsymbol{V}^T$, where $\boldsymbol{V}$ is the orthonormal matrix of the eigenvectors, and $\boldsymbol{\Lambda}$ is the diagonal matrix of corresponding eigenvalues. Note that we assume undirected graphs and thus $\boldsymbol{S}$ is symmetric. Spectral decomposition has been proven to be an efficient approach to generating meaningful node embedding for graph matching (Umeyama, 1988; Feizi et al., 2019). In particular, $\boldsymbol{E} = \boldsymbol{V}$ or $\boldsymbol{E} = \boldsymbol{V} \boldsymbol{\Lambda}$ are node embeddings that capture the network connectivity since they can perfectly reconstruct the graph. However, $\boldsymbol{V}$ is not unique. Thus computing the spectral decomposition of the same graph with node relabelling, $\tilde{\boldsymbol{S}} = \boldsymbol{P} \boldsymbol{S} \boldsymbol{P}^T$ is not guaranteed to produce a permuted version of $\boldsymbol{V}$, i.e., $\boldsymbol{P} \boldsymbol{V}$. Even in the case where $\boldsymbol{S}$ does not have repeated eigenvalues $\boldsymbol{V}$ is only unique up to column sign, which prevents effective matching.

To overcome the aforementioned uniqueness limitation, one can focus on the top $m$ eigenvectors that correspond to non-repeated eigenvalues in both $\boldsymbol{S}$ and $\hat{\boldsymbol{S}}$ and compute their absolute values. Then network alignment can be cast as:

$$\min_{\boldsymbol{P} \in \mathcal{P}} \ \left\| \ |\boldsymbol{V}_m| - \boldsymbol{P} \left| \hat{\boldsymbol{V}}_m \right| \ \right\|_F^2 , \tag{3}$$

where $\boldsymbol{V}_m \in \mathbb{R}^{N \times m}$ corresponds to the subspace of non-repeated eigenvalues. The formulation in (3) is a similar to the problem solved in (Umeyama, 1988).

# 3 Graph Neural Networks (GNNs) Upper-Bounds Spectral Methods for Network Alignment

A GNN is a cascade of layers and performs local, message-passing operations that are usually defined by the following recursive equation:

$$x_v^{(l+1)} = g\left(x_v^{(l)}, f\left(\left\{x_u^{(l)} : u \in \mathcal{N}(v)\right\}\right)\right) , \tag{4}$$

where $\mathcal{N}(v)$ is the neighborhood of vertex $v$, i.e., $u \in \mathcal{N}(v)$ iff $(u, v) \in \mathcal{E}$. The function $f$ operates on multisets ($\{\cdot\}$ represents a multiset) and $f$, $g$ are ideally injective. Common choices for $f$ are the summation or mean function, and for $g$ the linear function, or the multi-layer perceptron (MLP).

Overall, the output of the $L-$th layer of a GNN is a function $\phi(\boldsymbol{X}; \boldsymbol{S}, \mathcal{H}) : \mathbb{R}^{N \times D} \to \mathbb{R}^{N \times D_L}$, where $\boldsymbol{S}$ is the graph operator, and $\mathcal{H}$ is the tensor of the trainable parameters in all $L$ layers and produces $D_L-$ dimensional embeddings for the nodes of the graph defined by $\boldsymbol{S}$.

GNNs admit some very valuable properties. First, they are permutation equivariant:

**Theorem 3.1** ((Xu et al., 2019b; Maron et al., 2018)). *Let $\phi(\boldsymbol{X}; \boldsymbol{S}, \mathcal{H}) : \mathbb{R}^{N \times D} \to \mathbb{R}^{N \times D^L}$ be a GNN with parameters $\mathcal{H}$. For $\tilde{\boldsymbol{X}} = \boldsymbol{P}\boldsymbol{X}$ and $\tilde{\boldsymbol{S}} = \boldsymbol{P}\boldsymbol{S}\boldsymbol{P}^T$ that correspond to node relabelling according to the permutation matrix $\boldsymbol{P}$, the output of the GNN takes the form:*

$$\tilde{\boldsymbol{X}}^{(L)} = \phi\left(\tilde{\boldsymbol{X}}; \tilde{\boldsymbol{S}}, \mathcal{H}\right) = \boldsymbol{P}\phi(\boldsymbol{X}; \boldsymbol{S}, \mathcal{H}) \tag{5}$$

The above property is not satisfied by other spectral methods. GNNs are also stable (Gama et al., 2020), transferable (Ruiz et al., 2020), and have high expressive power (Xu et al., 2019b; Abboud et al., 2021; Kanatsoulis & Ribeiro, 2022).

### 3.1 GNNs and Network Alignment

To characterize the ability of a GNN to perform network alignment we first point out the GNNs perform nonlinear spectral operations. Details can be found in Appendix B. We can prove that:

**Theorem 3.2.** *Let $\mathcal{G}$, $\hat{\mathcal{G}}$ be graphs with adjacencies $\boldsymbol{S}$, $\hat{\boldsymbol{S}}$ that have non-repeated eigenvalues. Also let $\boldsymbol{P}^\diamond$, $\boldsymbol{P}^\dagger$ be solutions to the optimization problems in* (1) *and* (3) *respectively. Then there exists a GNN $\phi(\boldsymbol{X}; \boldsymbol{S}, \mathcal{H}) : \mathbb{R}^{N \times D} \to \mathbb{R}^{N \times D^L}$ such that:*

$$\left\| \boldsymbol{S} - \boldsymbol{P}^\diamond \hat{\boldsymbol{S}} \boldsymbol{P}^{\diamond T} \right\|_F^2 \leq \left\| \boldsymbol{S} - \boldsymbol{P}^* \hat{\boldsymbol{S}} \boldsymbol{P}^{*T} \right\|_F^2 \leq \left\| \boldsymbol{S} - \boldsymbol{P}^\dagger \hat{\boldsymbol{S}} \boldsymbol{P}^{\dagger T} \right\|_F^2, \tag{6}$$

*with*

$$\boldsymbol{P}^* = \underset{\boldsymbol{P} \in \mathcal{P}}{\arg\min} \left\| \phi(\boldsymbol{X}; \boldsymbol{S}, \mathcal{H}) - \boldsymbol{P}\phi\left(\hat{\boldsymbol{X}}; \hat{\boldsymbol{S}}, \mathcal{H}\right) \right\|_F^2 \tag{7}$$

The proof can be found in Appendix C. The assumption that the graph adjacencies have different eigenvalues is not restrictive. Real nonisomorphic graphs have different eigenvalues with very high probability (Haemers & Spence, 2004). Theorem 3.2 compares the network alignment power of a GNN with that of a spectral algorithm Umeyama (1988), that uses the absolute values of graph adjacency eigenvectors to match two different graphs. According to Theorem 3.2 there always exists a GNN that can perform at least as well as the spectral approach. The proof studies a GNN with white random input and measures the variance of the filter output. Then it shows that GNN layers are able to compute the absolute values of the graph adjacency eigenvectors when the adjacency has non-repeated eigenvalues. As a result there always exists a single layer GNN that outputs the same node features as the ones used in Umeyama (1988), which concludes our proof.

## 4 Proposed Method

We now leverage the favorable properties of GNNs (permutation equivariance, expressivity, and transferability) and design a GNN approach to tackle network alignment at a large-scale. Our approach learns low-dimensional node embeddings (Eq. 4) that enable graph matching via solving the linear assignment in (2) rather than a quadratic assignment problem in (1). In this section, we design a robust GNN framework such that the node embeddings are expressive to accurately match similar nodes and also stable to graph perturbations, so that they yield high-quality network alignment.

### 4.1 Learning geometry preserving embeddings

A fundamental property of node embeddings is to preserve the geometry and topological characteristics of the network. This will allow expressive node representations that can effectively approximate

the original problem in (1) with the problem in (2). To achieve this goal we leverage an auto-encoder architecture that reconstructs the original graph from the node embeddings. Results on GNN expressivity indicate that this reconstruction is doable under specific conditions (Abboud et al., 2021). To build topology-preserving embeddings we solve the following optimization problem:

$$\min_{\mathcal{H}} l\left(\rho\left(\phi\left(\boldsymbol{X};\boldsymbol{S},\mathcal{H}\right)\phi\left(\boldsymbol{X};\boldsymbol{S},\mathcal{H}\right)^T\right),\boldsymbol{S}\right), \tag{8}$$

where $l\left(\cdot\right)$ is the binary cross entropy (BCE) and $\rho\left(\cdot\right)$ is the logistic function.

## 4.2 LARGE-SCALE NODE REPRESENTATION LEARNING WITH GENERALIZED GRAPH AUTO-ENCODERS

The goal of the proposed framework is to learn a function that maps graphs to node representations and effectively match nodes from different graphs. This function is modeled by a GNN encoder $\phi\left(\boldsymbol{X};\boldsymbol{S},\mathcal{H}\right)$, where each layer is described by Eq. 4. The learned encoder should work for a family of training graphs $\{\mathcal{G}_0,\dots,\mathcal{G}_i,\dots,\mathcal{G}_I\}$ with a set of adjacency matrices $\mathbb{S}=\{\boldsymbol{S}_0,\dots,\boldsymbol{S}_i,\dots,\boldsymbol{S}_I\}$, rather than a single graph. So the idea is not to train an auto-encoder on a single graph but train a generalized graph auto-encoder by solving the following optimization problem.

$$\min_{\mathcal{H}} \mathbb{E}\left[l\left(\rho\left(\phi\left(\boldsymbol{X};\boldsymbol{S}_i,\mathcal{H}\right)\phi\left(\boldsymbol{X};\boldsymbol{S}_i,\mathcal{H}\right)^T\right),\boldsymbol{S}_i\right)\right], \tag{9}$$

where $\boldsymbol{S}_i \in \mathbb{S}$ is a realization from a family of graphs and the expectation (empirical expectation is practice) is computed over this graph family. The generalized framework in (9) learns a mapping from graphs to node representations, and can be applied to out-of-distribution graphs that have not been observed during training. This twist in the architecture enables node embedding and graph matching for large-scale graphs, where training is computationally prohibitive.

## 4.3 ROBUST AND GENERALIZABLE NODE REPRESENTATIONS WITH SELF-SUPERVISED LEARNING (DATA AUGMENTATION)

So far we proposed a convolutional framework to produce expressive node representations that are tailored to perform network alignment. In this subsection, we further upgrade our framework by ensuring the robustness and generalization ability of the proposed GNN mapping. In particular, for each graph, $\boldsymbol{S}_i \in \mathbb{S}$, we augment the training set with perturbed versions that are described by the following set of graph adjacencies $\mathbb{M}_i = \left\{\boldsymbol{S}_i^{(0)},\dots,\boldsymbol{S}_i^{(j)},\dots,\boldsymbol{S}_i^{(J)}\right\}$, that are perturbed versions of $\boldsymbol{S}_i$. To do so we add or remove an edge with a certain probability yielding $\tilde{\boldsymbol{S}}_i \in \mathbb{M}$, such that $\tilde{\boldsymbol{S}}_i = \boldsymbol{S}_i + \boldsymbol{M}_i$, where $\boldsymbol{M}_i \in \{-1,0,1\}^{N\times N}$. Note that $\boldsymbol{M}$ changes for each $\tilde{\boldsymbol{S}}_i$, and $\boldsymbol{M}[m,n]$ can be equal to $1$ and $-1$ only if $\boldsymbol{S}[m,n]$ is equal to $0$ and $1$ respectively. To train the proposed generalized graph-autoencoder we consider the following optimization problem:

$$\min_{\mathcal{H}} \mathbb{E}_{\mathbb{S}}\left[\mathbb{E}_{\mathbb{M}_i}\left[l\left(\rho\left(\phi\left(\boldsymbol{X};\tilde{\boldsymbol{S}}_i,\mathcal{H}\right)\phi\left(\boldsymbol{X};\tilde{\boldsymbol{S}}_i,\mathcal{H}\right)^T\right),\boldsymbol{S}_i\right)\right]\right], \tag{10}$$

where $\mathbb{E}_{\mathbb{S}}$ is the expectation with respect to the family of graphs $\mathbb{S}$ and $\mathbb{E}_{\mathbb{M}_i}$ is the expectation with respect to the perturbed graphs $\mathbb{M}_i$. In practice, $\mathbb{E}_{\mathbb{S}}$, $\mathbb{E}_{\mathbb{M}}$ correspond to empirical expectations. Note that training according to (10) also benefits the robustness of the model, which is crucial in deep learning tasks (Wang et al., 2022). A schematic illustration of the training process can be found in Fig. 1.

**Remark 4.1.** *(Large-scale network alignment by transference)*
*The proposed framework learns a mapping $\phi : \mathbb{G} \to \mathbb{R}^{N\times F}$ that produces expressive and robust node representations for a family of graphs $\mathcal{G} \in \mathbb{G}$. This mapping is designed in such a way that the problem in (2) approximates the problem in (1) and allows solving network alignment in polynomial time. One of the main benefits of the proposed framework is that it enables large-scale network alignment. The transferability analysis of GNN encoders (Ruiz et al., 2020), suggests that we can train with small graphs and efficiently execute with much larger graphs when the substructures (motifs) that appear in the tested graphs, were also partially observed during training. Since the proposed generalized graph auto-encoder is trained with multiple graphs, a variety of motifs are observed during training, which cannot be observed with a classical graph autoencoder, and the proposed GNN encoder can be transferred to large-scale graphs.*

| Task | Dataset | $|\mathcal{V}|$ | $|\mathcal{E}|$ | # Aligned Edges | Network Type |
|---|---|---|---|---|---|
| Graph Matching | Celegans (Kunegis, 2013) | 453 | 2,025 | 2,025 | Interactome |
| | Arenas (Leskovec & Krevl, 2014) | 1,135 | 3,982 | 3,982 | Email Communication |
| | Douban (Zhang & Tong, 2016) | 3,906 | 7,215 | 7,215 | Social Network |
| | Cora (Sen et al., 2008) | 2,708 | 5,278 | 5,278 | Citation Network |
| | Dblp (Pan et al., 2016) | 17,716 | 52,867 | 52,867 | Citation Network |
| | Coauthor CS (Shchur et al., 2018) | 18,333 | 81,894 | 81,894 | Coauthor Network |
| Subraph Matching | ACM-DBLP (Zhang & Tong, 2019) | 9,872 / 9,916 | 39,561 / 44,808 | 6,352 | Citation Network |
| | Douban Online-Offline (Zhang & Tong, 2016) | 3,906 / 1,118 | 1,632 / 3,022 | 1,118 | Social Network |

Table 2: Summary of Dataset statistics

## 4.4 ALIGNMENT AND COMPLEXITY ANALYSIS

After learning the powerful T-GAE node embeddings, network alignment is performed by solving the linear assignment problem in (2). An illustration of the assignment is presented in Fig. 2. The node features produced by T-GAE are used to calculate a pairwise distance matrix, followed by the greedy Hungarian algo rithm to predict node correspondences.

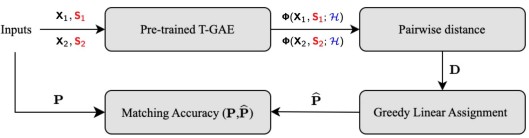

Figure 2: Alignment pipeline for T-GAE.

To analyze the complexity of our approach we study the 3 main parts of T-GAE: a) The design of the input structural features, b) The message-passing GNN that produces node embeddings, and c) the linear assignment algorithm. The computation of our neighborhood-based structural features is expected to take $\mathcal{O}\left(|\mathcal{V}|\right)$ in real graphs, as proved in Henderson et al. (2011). The computational and memory complexity of the message-passing GNN is $\mathcal{O}\left(|\mathcal{V}| c^2 + |\mathcal{E}| c\right)$, and $\mathcal{O}\left(|\mathcal{V}| c\right)$, where $c$ is the width of the GNN. The computational complexity to align the nodes of the graph is $\mathcal{O}\left(|\mathcal{V}|^2\right)$ since we are using the suboptimal greedy Hungarian. If we want to optimally solve the linear assignment problem we need to use the Hungarian algorithm that has $\mathcal{O}\left(|\mathcal{V}|^3\right)$ complexity. If we want to process large graphs we can embed the nodes in 1-dimensional space and use a sorting algorithm with complexity $\mathcal{O}\left(|\mathcal{V}| \log\left(|\mathcal{V}|\right)\right)$ to perform linear assignment. Overall the complexity of T-GAE is $\mathcal{O}\left(|\mathcal{V}|^2\right)$, or $\mathcal{O}\left(|\mathcal{V}| c^2 + |\mathcal{E}| c + |\mathcal{V}| \log\left(|\mathcal{V}|\right)\right)$ for large graphs.

## 5 EXPERIMENTS

In this section, we evaluate the performance of the proposed framework on both graph and sub-graph alignment with various benchmark networks. We compare against several baselines and assess the performance of the competing methods in terms of matching accuracy, hit-rate, and runtime.

## 5.1 DATASETS AND BASELINES

Table 2 provides a brief overview of the considered networks. Our comparisons are conducted with 3 categories of baseline methods: (a) **GNN based methods**: WAlign (Gao et al., 2021b), GAE and VGAE (Kipf & Welling, 2016a); (b) **Graph/Node embedding techniques**: NetSimile (Berlingerio et al., 2013), Spectral (Umeyama, 1988), DeepWalk (Perozzi et al., 2014), (Grover & Leskovec, 2016b), GraphWave (Donnat et al., 2018) and LINE (Tang et al., 2015). (c) **Optimization based graph matching algorithms**: S-GWL (Xu et al., 2019a), ConeAlign (Chen et al., 2020) and FINAL (Zhang & Tong, 2016). Note that LINE, VGAE, DeepWalk, and Node2Vec are omitted from some experiments since they show very poor performance. The reason behind that is that they are not permutation equivariant. GraphWave is also excluded from the sub-graph matching experiment, it could not identify correlated nodes in two different graphs. In the case of graphs without attributes FINAL is equivalent to the popular Isorank (Singh et al., 2008) algorithm. FINAL is omitted in sub-graph matching experiments due to weak performance.

## 5.2 MODEL DETAILS

For graph matching experiments, we consider graphs without node attributes, and design the input to the GNN models, using 7 structural features proposed in (Berlingerio et al., 2013). The features include the degree of each node, the local and average clustering coefficient, and the number of edges, outgoing edges, and neighbors in each node's egonet. This input feature is applied for all GNN-based methods. As a result, the performance of `NetSimile`, vanilla `GAE` and `WAlign` provide measures to assess the benefit of using `T-GAE` for node embedding.

As illustrated in Figure 1, the structure of our proposed encoder consists of two MLPs and a series of GNN layers. The node features are processed by a 2-layer MLP and passed to all the GNN layers. We add skip connections between this MLP layer and all the subsequent GNN layers. The outputs of all GNN layers are concatenated and passed to another 2-layer MLP, followed by a linear decoder to generate the reconstructed graph. The model is optimized end to end by equation 10. We test the performance of the proposed T-GAE framework by experimenting on three kinds of message-passing mechanisms on graphs, i.e., `GCN` (Kipf & Welling, 2016b), `GIN` (Xu et al., 2019b) and $GNN_c$ (described in Equation (11)). These mechanisms correspond to different functions $f$ and $g$ in Equation (4). We report the performance of `GIN` in the main body and the others in Appendix G.

## 5.3 GRAPH MATCHING EXPERIMENTS

To test the performance of the competing methods, we first attempt to match the graphs of Table 2 with permuted and perturbed versions of them. In particular, let $\mathcal{G}$ be a graph of Table 2 with adjacency matrix $\boldsymbol{S}$. For each graph we produce 10 permuted-perturbed versions according to $\hat{\boldsymbol{S}} = \boldsymbol{P}(\boldsymbol{S} + \boldsymbol{M})\boldsymbol{P}^T$, where $\boldsymbol{M} \in \{-1, 0, 1\}^{N \times N}$ and $\boldsymbol{P}$ is a permutation matrix. For each perturbation level $p \in \{0, 1\%, 5\%\}$, the total number of perturbations is defined as $p|\mathcal{E}|$, where $|\mathcal{E}|$ is the number of edges of the original graph. Then every edge and non-edge share the same probability of being removed or added. We also conducted experiments by removing edges according to the degrees of its vertices. Results for that model are discussed in Appendix H.

### 5.3.1 TRANSFERABILITY ANALYSIS

We first test the ability of `T-GAE` to perform large-scale network alignment and transfer across different datasets. To this end, we train `T-GAE` according to (9), where $\mathbb{S}$ consist of the small-size networks, i.e., Celegans, Arena, Douban, and Cora. Then we resort to transfer learning and use the `T-GAE` encoder to produce node embedding on (a) perturbed versions of Celegans, Arena, Douban, and Cora, and (b) larger graphs, i.e., Dblp, and Coauthor CS. Note that neither the larger graphs, nor the perturbed versions of the small graphs were considered during training. This is in contrast with all competing baselines that are retrained on every testing graph pair. The average and standard deviation of the matching accuracy for 10 randomly generated perturbation samples are presented in Table 3.

Our first observation is that for zero perturbation most algorithms are able to achieve a high level of matching accuracy. This is expected, since for zero perturbation the network alignment is equivalent to graph isomorphism. Furthermore, there is a clear benefit of processing the `NetSimile` embeddings with GNNs since they offer up to 22% performance increase. When some perturbation is added, the conclusions are straightforward. Our proposed `T-GAE` markedly outperforms all the competing alternatives and shows the desired robustness to efficiently perform network alignment at 1% perturbation level, and its performance is consistent across all datasets and perturbation levels.

Regarding the ability of `T-GAE` to perform large-scale network alignment the results are definitive. `T-GAE` enables low-complexity training with small graphs, and execution at larger settings by leveraging transfer learning. In particular, it is able to achieve very high levels of matching accuracy for both Dblp and Coauthor CS, for $p = 0\%$, $0.1\%$. To the best of our knowledge, this is the first attempt that performs exact alignment on a network at the order of 20k nodes and 80k edges.

Comparing T-GAE with vanilla GAE, we observe that GAE is not robust to noise or transferable. This highlights the benefit of T-GAE in handling the distribution shift brought by the structural dissimilarity between different graphs. We also notice that S-GWL completely fails the Arenas graph. This happens because Arenas has isolated nodes, and S-GWL struggles in handling such graphs. To see this, we also test S-GWL on the Arenas graph after removing all the isolated nodes and it achieves $94.6 \pm 0.5\%$ matching accuracy at 0 perturbation, $28.7 \pm 43.7\%$ matching accuracy

| | Dataset | Spectral | Netsimile | FINAL | ConeAlign | S-GWL | GraphWave | WAlign | GAE | **T-GAE** |
|---|---|---|---|---|---|---|---|---|---|---|
| **0% perturbation** | Celegans | 78.5 ± 18.4 | 72.7 ± 0.9 | 92.2 ± 1.2 | 66.6 ± 1.2 | **93.0 ± 1.5** | 65.3 ± 1.7 | 88.4 ± 1.6 | 80.2 ± 2.6 | 88.9 ± 1.3 |
| | Arenas | 87.9 ± 0.4 | 84.9 ± 0.8 | **88.1 ± 0.4** | 75.6 ± 0.4 | 0.1 ± 0.1 | 70.7 ± 1.3 | 88.3 ± 0.2 | 76.4 ± 1.1 | 87.8 ± 0.5 |
| | Douban | 82.1 ± 23.2 | 46.4 ± 0.4 | 89.9 ± 0.3 | 68.1 ± 0.4 | **90.1 ± 0.3** | 17.5 ± 0.2 | 90.0 ± 0.4 | 65.6 ± 0.4 | 89.9 ± 0.4 |
| | Cora | 85.0 ± 0.4 | 73.7 ± 0.4 | **87.5 ± 0.7** | 38.5 ± 0.7 | 87.3 ± 0.7 | 8.3 ± 0.4 | 87.2 ± 0.4 | 84.0 ± 0.8 | 87.3 ± 0.4 |
| | Dblp | 67.6 ± 33.8 | 63.7 ± 0.2 | **85.6 ± 0.2** | 44.3 ± 0.6 | > 10 hours | doesn't scale | 85.6 ± 0.2 | 74.1 ± 0.3 | **85.6 ± 0.2** |
| | Coauthor CS | 97.5 ± 0.1 | 90.9 ± 0.1 | **97.6 ± 0.1** | 75.8 ± 0.5 | > 10 hours | doesn't scale | 97.5 ± 0.2 | 71.1 ± 0.3 | **97.6 ± 0.1** |
| **1% perturbation** | Celegans | 68.5 ± 16.1 | 66.3 ± 3.8 | 33.2 ± 7.8 | 60.9 ± 2.5 | **92.4 ± 1.4** | 22.5 ± 22.4 | 80.7 ± 3.0 | 13.6 ± 11.9 | 84.1 ± 1.1 |
| | Arenas | 73.7 ± 8.4 | 80.1 ± 0.7 | 31.8 ± 8.7 | 73.2 ± 0.8 | 0.1 ± 0.1 | 39.0 ± 23.1 | 84.3 ± 0.5 | 9.0 ± 6.2 | **84.4 ± 1.2** |
| | Douban | 25.8 ± 27.2 | 40.0 ± 1.2 | 27.8 ± 5.7 | 64.7 ± 0.4 | 0.0 ± 0.0 | 9.9 ± 5.9 | 77.2 ± 4.8 | 3.3 ± 2.9 | **84.9 ± 0.6** |
| | Cora | 59.1 ± 9.3 | 66.4 ± 1.6 | 30.0 ± 3.3 | 33.5 ± 1.6 | 8.7 ± 26.0 | 3.7 ± 2.9 | 80.1 ± 1.2 | 12.5 ± 5.1 | **82.9 ± 0.5** |
| | Dblp | 55.6 ± 19.0 | 55.1 ± 1.7 | 15.2 ± 3.3 | 37.8 ± 1.1 | > 10 hours | doesn't scale | 73.1 ± 1.6 | 3.9 ± 0.9 | **79.1 ± 0.4** |
| | Coauthor CS | 58.2 ± 22.1 | 75.2 ± 2.2 | 13.3 ± 5.0 | 68.5 ± 2.8 | > 10 hours | doesn't scale | 75.2 ± 5.4 | 3.0 ± 1.2 | **86.5 ± 0.8** |
| **5% perturbation** | Celegans | 24.9 ± 15.9 | 41.1 ± 13.0 | 10.4 ± 2.7 | 50.5 ± 3.4 | **81.1 ± 27.0** | 7.6 ± 9.2 | 42.4 ± 21.1 | 3.9 ± 5.1 | 48.3 ± 9.1 |
| | Arenas | 44.0 ± 16.3 | 46.5 ± 7.6 | 8.0 ± 1.8 | **65.1 ± 1.1** | 0.0 ± 0.0 | 3.6 ± 3.6 | 30.0 ± 15.0 | 1.3 ± 0.8 | 47.1 ± 5.6 |
| | Douban | 23.8 ± 20.6 | 20.7 ± 4.6 | 7.8 ± 3.0 | 54.1 ± 1.2 | **0.0 ± 0.0** | 1.9 ± 2.8 | 21.9 ± 10.7 | 0.6 ± 0.6 | **57.9 ± 6.1** |
| | Cora | 29.5 ± 0.8 | 41.2 ± 3.3 | 6.7 ± 2.8 | 23.0 ± 2.0 | 16.3 ± 32.6 | 0.8 ± 0.3 | 33.4 ± 7.3 | 2.2 ± 0.9 | **58.2 ± 2.0** |
| | Dblp | 28.0 ± 7.8 | 19.5 ± 4.8 | 2.7 ± 0.9 | 24.4 ± 2.9 | > 10 hours | doesn't scale | 15.9 ± 8.3 | 0.5 ± 0.1 | **40.8 ± 2.1** |
| | Coauthor CS | 9.7 ± 5.0 | 26.3 ± 6.0 | 2.0 ± 0.4 | **51.4 ± 5.1** | > 10 hours | doesn't scale | 11.3 ± 7.5 | 0.1 ± 0.0 | 26.9 ± 5.4 |

Table 3: Graph matching accuracy on 10 randomly perturbed samples under different levels of edge editing. The proposed T-GAE is trained on the clean Celegans, Arena, Douban, and Cora networks, and tested on noisy versions of them and the larger Dblp, and Coauthor CS. We test 3 different message-passing mechanisms for the layers of T-GAE as annotated in the table. Accuracy above 80% is highlighted in green, 40% to 80% accuracy is in yellow, and performance below 40% is in red.

at 1% perturbation and 37.4 ± 45.8% accuracy at 5% perturbation. The performance of S-GWL is also unstable in different noise levels, as removing edges may result in graphs with isolated nodes. Detailed runtime comparisons between T-GAE and all competing methods are presented in Appendix F.

### 5.3.2 PERTURBED TRAINING

In the previous experiment, T-GAE is trained with a family of original graphs and tested to match perturbed versions of a larger family of graphs. T-GAE exhibited more robust performance compared to the baseline methods, however, its matching accuracy dropped significantly as the perturbation level of testing data increased. To tackle this problem we follow a self-supervised learning approach and train T-GAE with a family of real graphs and perturbations of them. We train according to (10), which aims to produce similar node embeddings for both the original graphs and perturbed versions of them. The data augmentation process follows the previously explained perturbation models. Similar to the previous experiment we train over the four small datasets and execute over all datasets. Note that training and testing is performed with different perturbations of the original graphs. Table 4 reports the testing results of the best T-GAE when training is performed with graph perturbations.

| Algorithm | Celegans | Arenas | Douban | Cora | Dblp | Coauthor CS |
|---|---|---|---|---|---|---|
| | | | 0% perturbation | | | |
| T-GAE | 89.5 ± 1.3 | 88.4 ± 0.5 | 90.3 ± 0.4 | 87.4 ± 0.4 | 85.6 ± 0.1 | 97.6 ± 0.1 |
| T-GAE with pert. | **89.7 ± 1.5** | **88.6 ± 0.6** | 90.1 ± 0.4 | **87.4 ± 0.5** | **85.7 ± 0.2** | **97.7 ± 0.1** |
| | | | 1% perturbation | | | |
| T-GAE | **84.1 ± 1.1** | 84.8 ± 0.6 | 84.9 ± 0.6 | 82.9 ± 0.5 | 79.1 ± 0.4 | 86.5 ± 0.8 |
| T-GAE with pert. | 83.4 ± 1.6 | **85.6 ± 0.5** | **85.2 ± 0.6** | **83.2 ± 0.9** | **79.7 ± 0.4** | **87.1 ± 1.0** |
| | | | 5% perturbation | | | |
| T-GAE | 50.8 ± 3.3 | 47.1 ± 5.6 | 57.9 ± 6.1 | **58.2 ± 2.0** | 40.8 ± 2.1 | 26.9 ± 5.4 |
| T-GAE with pert. | **52.3 ± 5.4** | **62.6 ± 2.2** | **58.5 ± 5.0** | 57.4 ± 2.7 | **43.6 ± 3.7** | **30.4 ± 7.4** |

Table 4: Performance Comparison of T-GAE when trained with/without perturbation.

We observe that incorporating graph perturbations in the training process significantly helps in high perturbation levels and benefits the robustness of the proposed method. On the other hand, when testing on low levels of perturbations, using the original graph or perturbations to train the T-GAE does not lead to significant changes. In particular, at 5% testing perturbation, T-GAE archives 15.5% increase in the Arenas dataset, whereas at 0% and 1% testing perturbation the increase is 0.2% and 0.8% respectively.

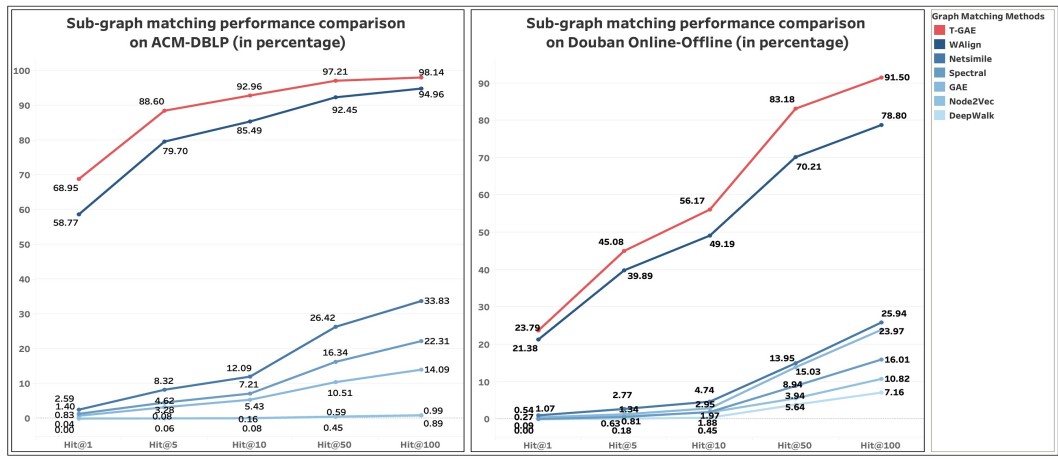

Figure 3: Subgraph Matching comparison between competing algorithms.

## 5.4 SUB-GRAPH MATCHING EXPERIMENTS

In this subsection, we test the performance of T-GAE in matching subgraphs of different networks that have aligned nodes (nodes that represent the same entities in different networks). For example, in ACM-DBLP data set, the task is to find and match the papers that appear in both citation networks, whereas in social networks like Douban Online-Offline, we aim to identify the users that take part into both online and offline activities. To this end, we test the performance of the proposed T-GAE framework on these datasets. We compare two different approaches. In the first, T-GAE is trained according to (9) to produce embedding for the graph pair we aim to match, i.e., the ACM-DBLP pair, or the Douban Online-Offline pair. In the second, T-GAE is trained according to (9) with Celegans, Arena, Douban, and Cora, and transfer learning is used to match the targeted graph pair. To assess the performance of the competing algorithms we measure the hit rate (Järvelin & Kekäläinen, 2000). The results are presented in Fig. 3. The execution time for the reported results is presented in Appendix F.

We observe a significant improvement in matching accuracy with GNN-based methods compared to traditional graph or node embedding techniques. These results demonstrate the ability of GNNs to generate expressive and robust node embeddings compared to classical algorithms. In particular, our proposed framework, T-GAE, consistently achieves the best performance among all competing methods. This suggests that the training framework (10), illustrated in Fig. 1, provides an efficient approach to network alignment. It is also notable, that T-GAE works well with both types of graph convolutions (GIN, GCN). This result indicates that the proposed framework has the potential to be extended to different types of neural networks.

Limitations: Although our approach achieves state-of-the-art performance in aligning real-graphs, approaching network alignment with a learning method, remains a heuristic and does not offer optimality guarantees. Furthermore, in order to process large graphs we cast network alignment as a self-supervised task. As a result in small-scale settings where the task can be tackled with computationally intensive efficient methods, our algorithm is not expected to perform the best. Finally, for large graphs the complexity of T-GAE $\mathcal{O}(|\mathcal{V}|^2)$ is limiting and therefore our alternative method with complexity $\mathcal{O}(|\mathcal{V}|c^2 + |\mathcal{E}|c + |\mathcal{V}|\log(|\mathcal{V}|))$ has to be employed.

## 6 CONCLUSION

We proposed T-GAE, a generalized transferable graph autoencoder to perform network alignment on a large scale. T-GAE can be trained with multiple graphs and produce robust and permutation equivariant embeddings tailored to network alignment. The produced embeddings are related to the spectral decomposition of the graph and are at least as good in graph matching as certain spectral methods. The proposed approach leverages transfer learning and data augmentation and achieves high levels of matching accuracy for graphs with more than $15,000$ nodes. Experiments with real-world benchmarks on both graph matching and subgraph matching tasks demonstrated the effectiveness and limits of the proposed approach.

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

## A  Notation and further reading

Our notation is summarized in Table 1.

| | | |
|---:|:---:|:---|
| $\mathcal{G}$ | $\triangleq$ | Graph |
| $\mathcal{V}$ | $\triangleq$ | Set of nodes |
| $\mathcal{E}$ | $\triangleq$ | Set of edges |
| $N$ | $\triangleq$ | Number of nodes |
| $\boldsymbol{D}$ | $\triangleq$ | Degree matrix |
| $\boldsymbol{S}$ | $\triangleq$ | $\{0,1\}^{N \times N}$ adjacency matrix |
| $\boldsymbol{X}$ | $\triangleq$ | $N \times D$ feature matrix |
| $\boldsymbol{H}$ | $\triangleq$ | aggregation results of GNN convolution |
| $\boldsymbol{W}$ | $\triangleq$ | weight matrix of the Graph Neural Network |
| $\boldsymbol{N}_v$ | $\triangleq$ | neighbors of node v |
| $\boldsymbol{I}$ | $\triangleq$ | Identity matrix |
| $\boldsymbol{0}$ | $\triangleq$ | vector or matrix of zeros |
| $\boldsymbol{A}^T$ | $\triangleq$ | transpose of matrix $\boldsymbol{A}$ |
| $\boldsymbol{A}_{rc}$ | $\triangleq$ | entry at r-th row and j-th column of matrix $\boldsymbol{A}$ |
| $\lVert \cdot \rVert_F$ | $\triangleq$ | Frobenius norm |

Table 1: Key notations used in this paper.

Graph matching, as an important problem, has been studied not only by the community of data mining, it has also been mathematically and statistically investigated. A number of approaches have been proposed to solve this problem for Erdős Rényi random graphs $G(n, \frac{d}{n})$. It has been proved that a perfectly true vertex correspondence can be recovered in polynomial time with high probability (Ding et al., 2020). Furthermore, a sharp threshold has been proved for both Erdős Rényi model and Gaussian model (Wu et al., 2022). Most recently, a novel approach to calculate similarity scores based on counting weighted trees rooted at each vertex has been proposed Mao et al. (2023). Such approach has been proved to be effective in solving the aforementioned network alignment problem on random graphs with high probability. Readers are encouraged to refer to the authors of these publications (Ding et al., 2020; Wu et al., 2022; Mao et al., 2023) for further reading.

## B  Spectral characterization of GNNs

What remains to be answered is the ability of a GNN to approximate a function that performs graph alignment. To understand the function approximation properties of GNNs we study them in the spectral domain. To this end, we consider the recursive formula in (4) where $f$ is the summation function and $g$ is multivariate linear for $K - 2$ layers, and the MLP in the $(K - 1)$-th layer. The overall operation can be written in a matrix form as:

$$\boldsymbol{X}^{(l+1)} = \sigma \left( \sum_{k=0}^{K-1} \boldsymbol{S}^k \boldsymbol{X}^{(l)} \boldsymbol{H}_k^{(l)} \right), \tag{11}$$

where $\boldsymbol{H}_k^{(l)} \in \mathbb{R}^{D^{l+1} \times D^l}$ is a linear mapping. Computing the spectral decomposition of $\boldsymbol{S}$ yields:

$$\boldsymbol{X}^{(l+1)} = \sigma \left( \sum_{k=0}^{K-1} \boldsymbol{V} \boldsymbol{\Lambda}^k \boldsymbol{V}^T \boldsymbol{X}^{(l)} \boldsymbol{H}_k^{(l)} \right) = \sigma \left( \sum_{k=0}^{K-1} \sum_{n=1}^{N} \lambda_n^k \boldsymbol{v}_n \boldsymbol{v}_n^T \boldsymbol{X}^{(l)} \boldsymbol{H}_k^{(l)} \right). \tag{12}$$

Then each each column of $\boldsymbol{X}^{(l+1)}$ can be written as

$$\boldsymbol{X}^{(l+1)}[:, i] = \sigma \left( \sum_{k=0}^{K-1} \sum_{n=1}^{N} \lambda_n^k \boldsymbol{v}_n \boldsymbol{v}_n^T \boldsymbol{X}^{(l)} \boldsymbol{H}_k^{(l)}[:, i] \right) = \sigma \left( \sum_{n=1}^{N} a_n^{(i)} \boldsymbol{v}_n \right), \tag{13}$$

where $\lambda_n, \boldsymbol{v}_n$ are the $n-$th eigenvalue and eigenvector and $a_n^{(i)} = \boldsymbol{v}_n^T \boldsymbol{X}^{(l)} \sum_{k=0}^{K-1} \lambda_n^k \boldsymbol{H}_k^{(l)}[:, i]$ is a scalar related to the Graph Fourier Transform (GFT) of $\boldsymbol{X}^{(l)}$ (Sardellitti et al., 2017). It is clear from

equation (13) that the output of each layer is a linear combination of the adjacency eigenvectors, followed by a pointwise non-linearity. Thus, a GNN can produce unique and more powerful graph embeddings than spectral methods by processing the eigenvectors and eigenvalues of the adjacency matrix.

## C  PROOF OF THEOREM 3.2

To prove Theorem 3.2. We consider one layer GNN with a vector input $\boldsymbol{x} \in \mathbb{R}^N$. This GNN can be represented by the following equation:

$$\boldsymbol{Y} = \sigma \left( \sum_{k=0}^{K-1} \boldsymbol{S}^k \boldsymbol{x} \boldsymbol{h}_k^T \right), \tag{14}$$

where $\boldsymbol{h}_k \in \mathbb{R}^m$ and $\boldsymbol{x}\boldsymbol{h}_k^T$ is an outer-product operation. The equation in (14) describes a set of $m$ graph filters of the form:

$$\boldsymbol{y}_i = \sigma \left( \sum_{k=0}^{K-1} h_k^i \boldsymbol{S}^k \boldsymbol{x} \right), \quad \text{for } i = 1, \dots, m \tag{15}$$

### C.1  WHITE RANDOM INPUT AND VARIANCE COMPUTATION

Let $\boldsymbol{x}$ be a white random vector with $\mathbb{E}\left[\boldsymbol{x}\right] = 0$ and $\mathbb{E}\left[\boldsymbol{x}\boldsymbol{x}^T\right] = \boldsymbol{I}$, where $\boldsymbol{I}$ is the diagonal matrix. Also let $\sigma\left(\cdot\right) = \left(\cdot\right)^2$ be the elementwise square function. Then (15) can be written as:

$$\boldsymbol{y}_i = \left( \sum_{k=0}^{K-1} h_k^i \boldsymbol{S}^k \boldsymbol{x} \right)^2 = \text{diag} \left( \sum_{k=0}^{K-1} h_k^i \boldsymbol{S}^k \boldsymbol{x}\boldsymbol{x}^T \sum_{j=0}^{K-1} h_j^i \boldsymbol{S}^{j^T} \right) \tag{16}$$

Since $\boldsymbol{x}$ is a random vector $\boldsymbol{y}_i$ is also a random vector. The expected value of $\boldsymbol{y}_i$ yields:

$$\mathbb{E}\left[\boldsymbol{y}_i\right] = \mathbb{E}\left[ \text{diag} \left( \sum_{k=0}^{K-1} h_k^i \boldsymbol{S}^k \boldsymbol{x}\boldsymbol{x}^T \sum_{j=0}^{K-1} h_j^i \boldsymbol{S}^{j^T} \right) \right] = \text{diag} \left( \sum_{k=0}^{K-1} h_k^i \boldsymbol{S}^k \mathbb{E}\left[\boldsymbol{x}\boldsymbol{x}^T\right] \sum_{j=0}^{K-1} h_j^i \boldsymbol{S}^{j^T} \right)$$

$$= \text{diag} \left( \sum_{k=0}^{K-1} h_k^i \boldsymbol{S}^k \sum_{j=0}^{K-1} h_j^i \boldsymbol{S}^{j^T} \right) \tag{17}$$

### C.2  SINGLE BAND FILTERING

In the second part of the proof we study the graph filter using the spectral decomposition of the graph:

$$\boldsymbol{y} = \sum_{k=0}^{K-1} h_k \boldsymbol{S}^k \boldsymbol{x} = \sum_{k=0}^{K-1} h_k \boldsymbol{V} \boldsymbol{\Lambda}^k \boldsymbol{V}^T \boldsymbol{x} = \sum_{k=0}^{K-1} h_k \sum_{n=1}^{N} \lambda_n^k \boldsymbol{v}_n \boldsymbol{v}_n^T \boldsymbol{x} = \sum_{n=1}^{N} \boldsymbol{v}_n^T \boldsymbol{x} \sum_{k=0}^{K-1} h_k \lambda_n^k \boldsymbol{v}_n. \tag{18}$$

Let us focus on the following polynomial:

$$\tilde{h}\left(\lambda\right) = \sum_{k=0}^{K-1} h_k \lambda^k, \tag{19}$$

that represents a graph filter in the frequency domain by. For $q$ distinct eigenvalues we can write a system of linear equations using the polynomial in (19):

$$\begin{bmatrix} \tilde{h}\left(\lambda_1\right) \\ \tilde{h}\left(\lambda_2\right) \\ \vdots \\ \tilde{h}\left(\lambda_q\right) \end{bmatrix} = \begin{bmatrix} 1 \ \lambda_1 \ \lambda_1^2 \dots \lambda_1^{K-1} \\ 1 \ \lambda_2 \ \lambda_2^2 \dots \lambda_2^{K-1} \\ \vdots \\ 1 \ \lambda_q \ \lambda_q^2 \dots \lambda_q^{K-1} \end{bmatrix} \begin{bmatrix} h_0 \\ h_1 \\ \vdots \\ h_{K-1} \end{bmatrix} = \boldsymbol{W}\boldsymbol{h} \tag{20}$$

$W$ is a Vandermonde matrix and when $K = q$ the determinant of $W$ takes the form:

$$\det\left(W\right) = \prod_{1 \leq i < j \leq q} \left(\lambda_i - \lambda_j\right) \tag{21}$$

Since the values $\lambda_i$ are distinct, $W$ has full column rank and there exists a graph filter with unique parameters $h$ that passes only the $\lambda$ eigenvalue, i.e.,

$$\tilde{h}\left(\lambda_i\right) = \begin{cases} 1, & \text{if } \lambda_i = \lambda \\ 0, & \text{if } \lambda_i \neq \lambda \end{cases} \tag{22}$$

Under this parametrization, equation (18) takes the form $y = v_\lambda v_\lambda^T x$, where $v_\lambda$ is the eigenvector corresponding to $\lambda$.

### C.3 GNN AND ABSOLUTE EIGENVECTORS

Using the previous analysis we can design parameters $h_k$ such that:

$$\sum_{k=0}^{K-1} h_k S^k = v_\lambda v_\lambda^T \tag{23}$$

and then equation (17) takes the form:

$$\mathbb{E}\left[y_i\right] = \text{diag}\left(\sum_{k=0}^{K-1} h_k^i S^k \sum_{j=0}^{K-1} h_j^i S^{j^T}\right) = \text{diag}\left(v_\lambda v_\lambda^T v_\lambda v_\lambda^T\right) = \text{diag}\left(v_\lambda v_\lambda^T\right) = |v_\lambda|^2 \tag{24}$$

We can therefore design $h_k \in \mathbb{R}^m$ for $k = 0, \ldots, m-1$ to compute the absolute value of $m$ eigenvectors of $S$ that correspond to the top $m$ distinct eigenvalues, i.e.,

$$\mathbb{E}\left[y_i\right] = |u_i|^2, \quad i = 1, \ldots, m \tag{25}$$
$$\tag{26}$$

We can do the same for graph $\hat{S}$ and compute:

$$\mathbb{E}\left[\hat{y}_i\right] = |\hat{u}_i|^2, \quad i = 1, \ldots, m \tag{27}$$
$$\tag{28}$$

Since both $S$, $\hat{S}$ have distinct eigenvalues, we can concatenate the output of each neuron and result in layer-1 outputs as:

$$Y^{(1)} = |U|, \quad \hat{Y}^{(1)} = |\hat{U}| \tag{29}$$

As a result, the previously described GNN can a least yield the same alignment accuracy as the absolute values of the eigenvectors.

## D DATA AUGMENTATION-PERTURBATION MODEL

Figure 4 gives an illustrative example of data augmentation process, used for training, and the perturbation model used during testing. In particular, we add or remove an edge with a certain probability yielding $S' = S + M$, where $M \in \{-1, 0, 1\}^{N \times N}$. Note that $M[i, j]$ can be equal to 1 and $-1$ only if $S[i, j]$ is equal to 0 and 1 respectively. Then we relabel each node by permuting the rows and columns of the adjacency matrix yielding $\hat{S} = P\left(S + M\right)P^T$.

## E IMPLEMENTATION DETAILS

In this section we discuss the implementation details of our framework. The code and datasets used in all experiments can be found here: `https://github.com/GraphMatching/Graph-Matching`

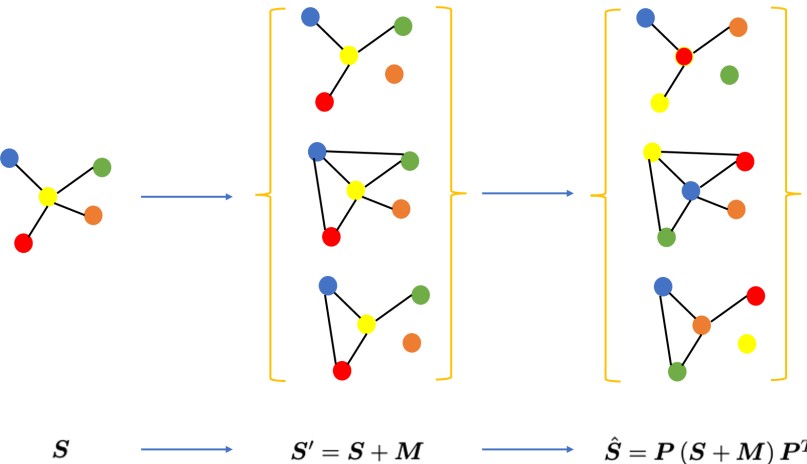

$$\boldsymbol{S} \quad\longrightarrow\quad \boldsymbol{S}' = \boldsymbol{S} + \boldsymbol{M} \quad\longrightarrow\quad \hat{\boldsymbol{S}} = \boldsymbol{P}\left(\boldsymbol{S} + \boldsymbol{M}\right)\boldsymbol{P}^T$$

Figure 4: The proposed data augmentation process. The first step is to remove existing edges and/or add new ones, while the second step reorders the nodes by permuting the rows and columns of the graph adjacency.

## E.1  ASSIGNMENT OPTIMIZATION

The proposed `T-GAE` learns learns a GNN encoder that can produce node representations for different graphs. Let $\phi\left(\boldsymbol{X};\boldsymbol{S},\mathcal{H}\right)$ represent the embeddings of the nodes corresponding to the graph with adjacency $\boldsymbol{S}$ and $\phi\left(\hat{\boldsymbol{X}};\hat{\boldsymbol{S}},\mathcal{H}\right)$ represent the embeddings of the nodes corresponding to the graph with adjacency $\hat{\boldsymbol{S}}$. Then network alignment boils down to solving the following optimization problem:

$$\min_{\boldsymbol{P}\in\mathcal{P}} \left\| \phi\left(\boldsymbol{X};\boldsymbol{S},\mathcal{H}\right) - \boldsymbol{P}\phi\left(\hat{\boldsymbol{X}};\hat{\boldsymbol{S}},\mathcal{H}\right) \right\|_F^2. \tag{30}$$

The problem in (30) can be optimally solved in $\mathcal{O}\left(N^3\right)$ flops by the Hungarian algorithm (Kuhn, 1955a). To avoid this computational burden we employ the greedy Hungarian approach that has computational complexity $\mathcal{O}\left(N^2\right)$ and usually works well in practice.

The greedy Hungarian approach is described in Algorithm 1. For each row of $\phi\left(\boldsymbol{X};\boldsymbol{S},\mathcal{H}\right),\ \phi\left(\hat{\boldsymbol{X}};\hat{\boldsymbol{S}},\mathcal{H}\right)$, which corresponds to the node embeddings of the different graphs, we compute the pairwise Euclidean distance which is stores in the distance matrix $\boldsymbol{D}$. Then, at each iteration, we find the nodes with the smallest distance and remove the aligned pairs from $\boldsymbol{D}$. This process is repeated until all the nodes are paired up for alignment.

## E.2  DATASETS

The descriptions of the real-world datasets used in our experiments are presented below:

- Celegans (Kunegis, 2013): The vertices represent proteins and the edges their protein-protein interactions.

- Arenas Email (Leskovec & Krevl, 2014): The email communication network at the University Rovira i Virgili in Tarragona in the south of Catalonia in Spain. Nodes are users and each edge represents that at least one email was sent.

- Douban (Zhang & Tong, 2016): Contains user-user relationship on the Chinese movie review platform. Each edge implies that two users are contacts or friends.

---

**Algorithm 1:** Greedy Hungarian Algorithm

---

**Input:** Feature matrices $X, \hat{X}$
**Output:** Assignment Matrix

1  $P := \mathbf{0}_{N \times N}$                                  `// Initialize permutation matrix`

2  $D := \text{PairwiseDistance}\left(X, \hat{X}\right)$            `// pairwise Euclidean distance`

3  rows := 0,1,…,$N$-1                           `// Corresponds to` $X$

4  cols := 0,1,…,$N$-1                           `// Corresponds to` $\hat{X}$

    `/* Iterate to assign node pairs with minimum Euclidean distance     */`

5  **for** $n=1$ to $N$ **do**

6      $i, j := \text{argmin}\,(D)$

7      $r := \text{rows}\,[i]$

8      $c := \text{cols}\,[j]$

9      $P_{rc} := 1$

10     Remove $r$ from rows

11     Remove $c$ from cols

12     Remove the $i$-th row from $D$

13     Remove the $j$-th column from $D$

14  return $P$

---

- Cora (Sen et al., 2008): The dataset consists of 2708 scientific publications, with edges representing citation relationships between them. Cora has been one of the major benchmark datasets in many graph mining tasks.

- Dblp (Pan et al., 2016): A citation network dataset that is extracted from DBLP, Association for Computing Machinery (ACM), Microsoft Academic Graph (MAG), and other sources. It is considered a benchmark in multiple tasks.

- Coauthor_CS (Shchur et al., 2018): The coauthorship graph is generated from MAG. Nodes are the authors and they are connected with an edge if they coauthored at least one paper.

- ACMDBLP (Zhang & Tong, 2019): The citation networks that share some common nodes. The task is to identify the publications that appear in both networks.

- Douban OnlineOffline (Zhang & Tong, 2016): The two social networks contained in this dataset represents the online and offline events of the Douban social network. The task is to identify users that participate in both online and offline events.

### E.3 BASELINES

#### E.3.1 GRAPH NEURAL NETWORK(GNN) BASED METHODS

To have a fair comparision with the node embedding models, all GNN methods are set to have the hidden dimension equal to 4 in the graph matching experiments. In sub graph matching, the hidden dimension of GNNS are fine-tuned from 4 to 512. Other parameters are the same as the author suggested.

- WAlign (Gao et al., 2021b) fits a GNN to each of the input graphs, trains the model by reconstructing the given inputs and minimizing an approximation of Wasserstein distance between the node embeddings. We use the author's implementation from https://github.com/gaoji7777/walign.git.

- GAE, VGAE(Kipf & Welling, 2016a) are self-supervised graph learning frameworks that are trained by reconstructing the graph. The encoder is a GCN(Kipf & Welling, 2016b) and linear decoder is applied to predict the original adjacency. In VGAE, Gausian Noise is introduced to the node embeddings before passing to the decoder. We use the implementation from https://github.com/DaehanKim/vgae_pytorc.

### E.3.2 GRAPH/NODE EMBEDDING TECHNIQUES

- `NetSimile` (Berlingerio et al., 2013) uses the structural features described earlier to match the nodes of the graphs. Since the `NetSimile` features are used as input to the `T-GAE`, they provide a measure to assess the benefit of using `T-GAE` for node embedding. It proposed 7 egonet-based features, to measure network similarity. We process these features by Algorithm 1 to perform network alignment. The 7-dimensional Netsimile features are:
  - $d_i$ = degree of node i
  - $c_i$ = number of triangles connected to node i over the number of connected triples centered on node i
  - $\bar{d}_{N_i} = \frac{1}{d_i} \sum_{j \in N_i} d_j$, average number of two-hop neighbors
  - $\bar{c}_{N_i} = \frac{1}{d_i} \sum_{j \in N_i} c_j$, average clustering coefficient
  - Number of edges in node i's egonet
  - Number of outgoing edges from node i's egonet
  - Number of neighbors in node i's egonet

  The implementation is based on netrd library where we use the feature extraction function. The source code can be found at `https://netrd.readthedocs.io/en/latest/_modules/netrd/distance/netsimile.html`

- `Spectral` (Umeyama, 1988) It solves the following optimization problem:

$$\min_{\boldsymbol{P} \in \mathcal{P}} \; \left\| \; |\boldsymbol{V}| - \boldsymbol{P} \left| \hat{\boldsymbol{V}} \right| \; \right\|_F^2, \tag{31}$$

  where $\boldsymbol{V}$, $\hat{\boldsymbol{V}}$ are the eigenvectors corresponding to the adjacencies of the graphs that we want to match. In our initial experiments, we observed that a subset of the eigenvectors yields improved results compared to the whole set. We tried $1 - 10$ top eigenvectors and concluded that $4$ eigenvectors are those that yield the best results on average. Thus we solve the above problem with the top-4 eigenvectors.

- `DeepWalk` (Perozzi et al., 2014): A node embedding approach, simulates random walks on the graph and apply skip-gram on the walks to generate node embedding. We use the implementation from Karateclub `https://github.com/benedekrozemberczki/karateclub/blob/master/karateclub/node_embedding/neighbourhood/deepwalk.py` The algorithm is implemented with the default parameters as suggested by this repository, the number of random walks is 10 with each walk of length 80. The dimensionality of embedding is set to be 128. We run the algorithm with 1 epoch and set the learning rate to be 0.05.

- `Node2Vec` (Grover & Leskovec, 2016b): An improve version of DeepWalk, it has weights on the randomly generated random walks, to make the neighborhood preserving objective more flexible. We use the implementation from `https://github.com/benedekrozemberczki/karateclub/blob/master/karateclub/node_embedding/neighbourhood/node2vec.py`. The default parameters are used. We simulate 10 random walks on the graph with length 80. p and q are both equal to 1. Dimensionality of embeddings is set to be 4 and we run 1 epoch with learning rate 0.05.

- `GraphWave` (Donnat et al., 2018): The structure information of the graphs is captured by simulating heat diffusion process on them. We use the implementation from (Donnat et al., 2018):`https://github.com/benedekrozemberczki/karateclub/blob/master/karateclub/node_embedding/structural/graphwave.py` with the default parameters: number of evaluation points is 200, step size is 0.1, heat coefficient is 1.0 and Chebyshev polynomial order is set to be 100. Note that this implementation does not work on graphs with more than 10,000 nodes, so we exclude this model on the DBLP and Coauthor_CS dataset.

- LINE (Tang et al., 2015): An optimization based graph embedding approach that aims to preserve local and global structures of the network by considering substructures and structural-quivariant nodes. We use the PyTorch implementation from `https://github.com/zxhhh97/ABot`. All parameters are set to default as the authors suggested.

### E.3.3 OPTIMIZATION BASED GRAPH MATCHING ALGORITHMS

- FINAL (Zhang & Tong, 2016) is an optimization approach, following an alignment consistency principle, and tries to match nodes with similar topology. In the case of graphs without attributes FINAL is equivalent to the popular

- Isorank (Singh et al., 2008) algorithm, whereas using NetSimile as an input to FINAL resulted in inferior performance and was therefore omitted. We use the code in `https://github.com/sizhang92/FINAL-KDD16` with $H$ being the degree similarity matrix, $\alpha = 0.8$, maxiter = 30, tol = $1e - 4$ as suggested in the repository.

- ConeAlign (Chen et al., 2020) is a graph embedding based approach. The matching is optimized in each iteration by the Wasserstein Procrustes distances between the matched embeddings calculated on a mini batch in order to preserve scalability. We use the official implementation from `https://github.com/GemsLab/CONE-Align` and preserved all the suggested parameters.

- S-GWL (Xu et al., 2019a) matches two given graphs by retrieving node correspondence from the optimal transport associated with the Gromov-Wasserstein discrepancy between the graphs. We use the implementation by the authors in `https://github.com/HongtengXu/s-gwl` with all the suggested parameters.

### E.4 T-GAE PARAMETERS

We implement our proposed Transferable graph autoencoder (T-GAE) framework, following Equations (8), (9), (10), for the tasks described later in the section. We use 1 MLP and 6 GNN layers, followed by another MLP to design the Encoder $\phi(\boldsymbol{X}, \boldsymbol{S}; \mathcal{H})$ of T-GAE, as illustrated in Figure 1. For the GNN layers we use 3 different models: i) graph convolutional network (GCN) (Kipf & Welling, 2017), which is a message-passing architecture, described in Equation (4) with $f$ being the mean function, and $g$ being the linear function, ii) graph isomorphism network (GIN) (Xu et al., 2019b), which is also described by (4) with $f$ being the summation function, and $g$ being the MLP, and iii) graph neural network with k-hop neighborhood convolutions (GNN$_c$), as described by Equation (11). For graph matching experiments, in order to run experiments on large scale networks, we use an output dimension of 4. On sub-graph matching tasks, we fine tune this parameter from 4 to 512 and used the one that gives the best performance.

To select the hyperparameters of our model we utilize a validation graph (JAZZ (Rossi & Ahmed, 2015)), which is not included in our comparisons to ensure unbiased comparisons. Note that all parameters of T-GAE are fixed for all datasets that are reported. The model is not fine-tuned on every network that is used in testing.

The parameters of T-GAE include:

- We train our model Adam optimizer for 100 epochs and initial learning rate equal to 0.001 for graph matching experiment. To have a fair comparision with WAlign (Gao et al., 2021b), we train 20 epochs for the subgraph matching tasks.

- The hidden dimension of the GNN modules is equal to 6 and that of the output layer (encoder) is equal to 4 for graph matching experiments. For subgraph matching, the hidden dimensions are set to be 512, the same to the optimal value for WAlign (Gao et al., 2021b).

- The activation function for each GNN layer is ReLU.

- We use an 1-layer decoder followed by the sigmoid function.

- To conduct the experiments in Sections 5.3.2, G.3 we augment the dataset with 25 perturbed graphs for each perturbation level.

## F EFFICIENCY AND RUN-TIME COMPARISON

We analyze the efficiency of the proposed graph matching framework by comparing its running time with the competing algorithms. The GNN-based methods achieve at most $\times 50$ less running time on both graph matching and subgraph matching tasks. The proposed training objective (10) scales well

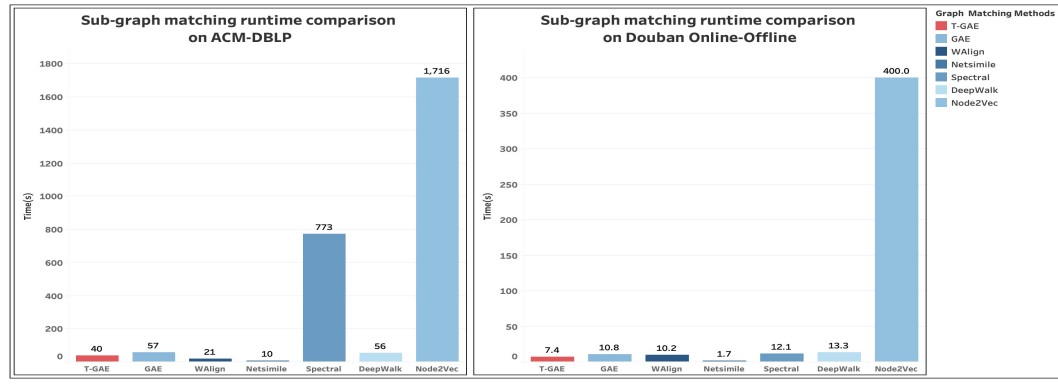

Figure 5: Runtime comparison for the competing algorithms on the sub-graph matching datasets. All GNN-based methods have 8 layers with 512 as hidden dimensions and are trained with 20 epochs.

from networks with 1,000 nodes (Zhang & Tong, 2016) to denser networks with $\times 10$ nodes, with minor running time increase, compared to other node embedding techniques.

We also noticed that compared to the other GNN based framework WAlign Gao et al. (2021b), T-GAE takes more time to train on the larger scale networks as illustrated in Figure 5. This is because the deep GNN is trained by the structure of the network. The training takes more time as the netowrk scales up and the number of nodes and edges grow dramatically, whereas the training labels for WAlign come from the input feature, which does not scale up with as the size of the training graphs increase. This could be one of the limitations to the proposed training paradigm and that is why we leverage transfer learning to efficiently match large scale graphs, as illustrated in Table3.

| Algorithm | Celegans | Arenas | Douban | Cora | Dblp | Coauthor CS |
|---|---|---|---|---|---|---|
| T-GAE | 0.098 | 0.241 | 1.934 | 0.842 | 173.836 | 193.289 |
| GAE | 0.094 | 0.252 | 1.949 | 0.857 | 174.410 | 194.062 |
| WAlign | 0.078 | 0.205 | 1.800 | 0.766 | 169.694 | 189.032 |
| Spectrul | 5.712 | 2.819 | 60.298 | 54.770 | > 10 hours | > 10 hours |
| GraphWave | 8.308 | 32.994 | 281.629 | 131.230 | 5470.724 | 6291.368 |
| ConeAlign | 1.333 | 3.500 | 31.955 | 13.799 | 887.090 | 1099.145 |
| S-GWL | 27.844 | 37.443 | 3394.522 | 311.201 | > 10 hours | > 10 hours |
| FINAL | 0.030 | 0.081 | 1.007 | 0.498 | 86.788 | 118.065 |

Table 2: Runtime in seconds for the competing algorithms on graph matching tasks.

# G  UNIFORM PERTURBATION MODEL SUPPLEMENTARY RESULTS

## G.1  GRAPH MATCHING RESULTS FOR T-GAE USING GCN AND GNN$_c$ AS BACKBONES

We present the graph matching accuracy of the proposed T-GAE method using different message passing mechanisms as the backbone of the encoder in Table 3. The experiment setup is the same as in Table 3, we train on the 4 smaller datasets (Celegans, Arenas, Douban, Cora) and test on 10 randomly generated perturbations of all the 6 datasets. We observe that the proposed framework work well with different types of message passing GNNs in graph matching tasks and is able to produce robust node embedding at small perturbation levels.

## G.2  MORE BASELINE RESULTS

We present the graph matching accuracy for the baseline methods that are not permutation equavariant in Table 4, and sub-graph matching accuracy on Douban Online-Offline dataset for GraphWave in Table 5, as it is not scalable on ACM/DBLP.

| | Algorithm | Celegans | Arenas | Douban | Cora | Dblp | Coauthor CS |
|---|---|---|---|---|---|---|---|
| 0% | T-GAE (GNN$_c$) | $89.5 \pm 1.3$ | $88.4 \pm 0.5$ | $90.1 \pm 0.4$ | $87.1 \pm 0.6$ | $85.6 \pm 0.1$ | $97.6 \pm 0.1$ |
| 0% | T-GAE (GCN) | $87.8 \pm 1.1$ | $88.3 \pm 0.4$ | $89.9 \pm 0.2$ | $87.4 \pm 0.4$ | $85.8 \pm 0.1$ | $97.6 \pm 0.1$ |
| 1% | T-GAE (GNN$_c$) | $83.4 \pm 1.6$ | $83.3 \pm 1.0$ | $83.3 \pm 0.6$ | $81.0 \pm 0.9$ | $75.3 \pm 1.5$ | $81.6 \pm 1.1$ |
| 1% | T-GAE (GCN) | $83.4 \pm 2.3$ | $84.8 \pm 0.6$ | $84.1 \pm 1.2$ | $82.5 \pm 0.8$ | $76.0 \pm 1.1$ | $84.3 \pm 0.8$ |
| 5% | T-GAE (GNN$_c$) | $50.8 \pm 3.3$ | $45.2 \pm 8.0$ | $44.5 \pm 7.5$ | $47.3 \pm 3.3$ | $32.7 \pm 1.6$ | $21.7 \pm 5.2$ |
| 5% | T-GAE (GCN) | $37.5 \pm 9.7$ | $45.0 \pm 8.2$ | $49.5 \pm 4.2$ | $50.9 \pm 2.8$ | $31.8 \pm 2.8$ | $23.0 \pm 4.1$ |

Table 3: Graph matching accuracy of the proposed T-GAE using GCN and GNN$_c$ as backbones

| | Dataset | VGAE | LINE | DeepWalk |
|---|---|---|---|---|
| 0% | Celegans | $0.3 \pm 0.1$ | $1.0 \pm 0.5$ | $1.8 \pm 0.6$ |
| 0% | Arenas | $0.1 \pm 0.1$ | $0.2 \pm 0.1$ | $0.3 \pm 0.2$ |
| 0% | Douban | $0.0 \pm 0.0$ | $0.0 \pm 0.0$ | $0.1 \pm 0.0$ |
| 0% | Cora | $0.1 \pm 0.0$ | $0.0 \pm 0.0$ | $0.1 \pm 0.0$ |
| 1% | Celegans | $0.3 \pm 0.1$ | $1.0 \pm 0.4$ | $1.2 \pm 0.5$ |
| 1% | Arenas | $0.1 \pm 0.1$ | $0.1 \pm 0.1$ | $0.3 \pm 0.1$ |
| 1% | Douban | $0.0 \pm 0.0$ | $0.0 \pm 0.0$ | $0.1 \pm 0.0$ |
| 1% | Cora | $0.1 \pm 0.1$ | $0.1 \pm 0.0$ | $0.2 \pm 0.1$ |
| 5% | Celegans | $0.6 \pm 0.3$ | $0.9 \pm 0.3$ | $1.0 \pm 0.3$ |
| 5% | Arenas | $0.2 \pm 0.1$ | $0.2 \pm 0.2$ | $0.2 \pm 0.1$ |
| 5% | Douban | $0.0 \pm 0.0$ | $0.0 \pm 0.0$ | $0.0 \pm 0.0$ |
| 5% | Cora | $0.1 \pm 0.0$ | $0.1 \pm 0.0$ | $0.1 \pm 0.0$ |

Table 4: Graph matching accuracy on 10 randomly perturbed samples under different levels of edge editing for VGAE, LINE and DeepWalk

### G.3    SPECIFIC PERTURBED TRAINING

The last question that we will try to answer with our experiments about Uniform Perturbation model is whether training and testing on the same graph compares to training with a family of graphs or a combination of a family and perturbed graphs, that was examined in the previous subsection. In this subsection we test the performance of `T-GAE` on Celegans, Arenas, Douban, and Cora, on 4 different settings: i) when training on each graph separately according to (8); ii) when training on all 4 graphs according to (9); iii) when training on the perturbations of only one graph according to (9), but $\mathbb{S}$ being defined over a family of graph perturbations; iv when training on all 4 graphs plus perturbed versions of them, according to (10). The results for the different levels of perturbations are presented in Table 6.

Table 6 shows when testing on low perturbation or no perturbations, all the models work statistically the same. The only exception is in the Celegans dataset where training on a family of graphs seems to improve the matching performance by $2.8\%$ compared to training just on a single graph. For $5\%$ perturbation level, on the other hand, either `T-GAE specific with pert.` or `T-GAE with pert.` yield the best results, which implies that a self-supervised approach with perturbations benefits performance and robustness. We also observe that `T-GAE with pert.`, i.e., training according to (10) is in general the most consistent approach.

## H    DEGREE PERTURBATION MODEL RESULTS

**Degree Model:** In this model we only remove edges. Edges with higher degrees are more likely to be removed to preserve the structure of the graph. Specifically, the probability of removing edge $(i, j)$ is set to $\frac{s_{ij} d_i d_j}{\sum_{ij} s_{ij} d_i d_j}$, where $d_i$ is the degree of node $v_i$ and $s_{i,j}$ is the $(i, j)$ element of the graph adjacency.

| | Hit rate | | GraphWave |
|---|---|---|---|
| | Hit@1 | | 0.09 |
| | Hit@5 | | 0.36 |
| | Hit@10 | | 0.81 |
| | Hit@50 | | 4.74 |
| | Hit@100 | | 9.12 |

Table 5: Sub-graph matching performance for GraphWave on Douban Online-Offline

| Perturbation | Celegans | Arenas | Douban | Cora |
|---|---|---|---|---|
| 0% perturbation | | | | |
| T-GAE specific (8) | $89.3 \pm 1.4$ | $88.3 \pm 0.4$ | $90.2 \pm 0.5$ | $87.5 \pm 0.4$ |
| T-GAE (9) | $89.5 \pm 1.3$ | $88.4 \pm 0.5$ | $90.1 \pm 0.4$ | $87.4 \pm 0.4$ |
| T-GAE specific with pert. | $\mathbf{90.4 \pm 0.8}$ | $88.5 \pm 0.4$ | $\mathbf{90.2 \pm 0.3}$ | $\mathbf{87.6 \pm 0.5}$ |
| T-GAE with pert. (10) | $89.7 \pm 1.5$ | $\mathbf{88.6 \pm 0.6}$ | $90.1 \pm 0.4$ | $87.4 \pm 0.5$ |
| 1% perturbation | | | | |
| T-GAE specific (8) | $81.3 \pm 1.7$ | $85.4 \pm 0.5$ | $85.1 \pm 1.1$ | $83.1 \pm 1.0$ |
| T-GAE (9) | $\mathbf{84.1 \pm 1.1}$ | $84.8 \pm 0.6$ | $84.9 \pm 0.6$ | $82.9 \pm 0.5$ |
| T-GAE specific with pert. | $83.1 \pm 1.4$ | $85.3 \pm 0.9$ | $\mathbf{85.3 \pm 0.7}$ | $\mathbf{83.5 \pm 0.6}$ |
| T-GAE with pert. (10) | $83.4 \pm 1.6$ | $\mathbf{85.6 \pm 0.5}$ | $85.2 \pm 0.6$ | $83.2 \pm 0.9$ |
| 5% perturbation | | | | |
| T-GAE specific (8) | $48.9 \pm 9.5$ | $61.5 \pm 5.4$ | $58.0 \pm 3.9$ | $58.8 \pm 2.7$ |
| T-GAE (9) | $50.8 \pm 3.3$ | $47.1 \pm 5.6$ | $57.9 \pm 6.1$ | $58.2 \pm 2.0$ |
| T-GAE specific with pert. | $\mathbf{53.5 \pm 11.9}$ | $54.3 \pm 6.4$ | $57.3 \pm 4.6$ | $\mathbf{58.9 \pm 2.4}$ |
| T-GAE with pert. (10) | $52.3 \pm 5.4$ | $\mathbf{62.6 \pm 2.2}$ | $\mathbf{58.5 \pm 5.0}$ | $57.4 \pm 2.7$ |

Table 6: Performance of T-GAE when trained on a family of graphs or on a specific graph on uniform perturbation model

## H.1 TRANSFERABILITY ANALYSIS

We test the performance of `T-GAE` for large-scale network alignment on the degree perturbation model, as described in Section 5.3. We adopt the same setting as in Section 5.3 to train the `T-GAE` according to (9) on small-size networks, i.e., Celegans, Arena, Douban, and Cora, and conduct transfer learning experiments on the larger graphs, i.e., Dblp, and Coauthor CS. The trained T-GAE is used to generate node embedding for the graphs, and 1 computes the assignment matrix. The results presented in Table 7 are based on the average and standard deviation of the matching accuracy over 10 randomly generated perturbed samples.

When testing on 0 perturbation level most algorithms are able to discover the node correspondences, as in the uniform perturbation model. The benefit of processing the `NetSimile` embeddings with GNNs is still significant in this perturbation model as we observe up to 53% performance increase. When testing on perturbed graphs, our proposed `T-GAE` consistently outperforms all the competing baselines, while being robust and efficient when performing network alignment under different perturbation models. Finally, on large-scale networks, `T-GAE` is able to achieve very high levels of matching accuracy for both Dblp and Coauthor CS, for $p = 0\%, 1\%$.

## H.2 PERTURBED TRAINING

We train the `T-GAE` with a family of original graphs and test to match perturbed versions of a larger family of graphs. We notice that although being more robust to perturbations than the baseline methods, the matching accuracy of `T-GAE` dropped as the perturbation level of testing data increased, especially on the degree perturbation model. To tackle this problem we follow a self-supervised learning approach and train `T-GAE` with a family of real graphs and perturbations of them. We train according to (10), which encourages the model to produce similar node embeddings for the original graphs and the perturbed versions of them. We augment the dataset as illustrated in 4, adding

| | Algorithm | Celegans | Arenas | Douban | Cora | Dblp | Coauthor CS |
|---|---|---|---|---|---|---|---|
| **0% perturbation** | Spectral | $66.5 \pm 28.4$ | $87.8 \pm 0.5$ | $35.7 \pm 35.5$ | $84.7 \pm 0.5$ | $84.6 \pm 0.1$ | $\mathbf{97.6 \pm 0.1}$ |
| | NetSimile | $71.8 \pm 1.0$ | $84.7 \pm 0.5$ | $46.9 \pm 0.5$ | $73.0 \pm 0.7$ | $63.7 \pm 0.2$ | $90.1 \pm 0.1$ |
| | FINAL | $\mathbf{92.7 \pm 1.9}$ | $88.2 \pm 0.5$ | $90.0 \pm 0.4$ | $87.1 \pm 0.5$ | $\mathbf{85.6 \pm 0.1}$ | $\mathbf{97.6 \pm 0.1}$ |
| | T-GAE (GNN$_c$) | $89.3 \pm 1.2$ | $88.1 \pm 0.4$ | $\mathbf{90.2 \pm 0.4}$ | $87.5 \pm 0.5$ | $\mathbf{85.6 \pm 0.1}$ | $\mathbf{97.6 \pm 0.1}$ |
| | T-GAE (GCN) | $89.7 \pm 1.4$ | $88.2 \pm 0.5$ | $90.1 \pm 0.3$ | $87.4 \pm 0.4$ | $\mathbf{85.6 \pm 0.1}$ | $\mathbf{97.6 \pm 0.1}$ |
| | T-GAE (GIN) | $89.6 \pm 1.0$ | $\mathbf{88.2 \pm 0.4}$ | $90.1 \pm 0.3$ | $\mathbf{87.6 \pm 0.4}$ | $85.4 \pm 0.1$ | $97.5 \pm 0.1$ |
| **1% perturbation** | Spectral | $21.8 \pm 16.9$ | $62.9 \pm 21.8$ | $9.8 \pm 13.5$ | $25.3 \pm 13.4$ | $3.4 \pm 0.8$ | $8.6 \pm 4.6$ |
| | NetSimile | $63.1 \pm 1.4$ | $81.1 \pm 0.7$ | $37.6 \pm 0.6$ | $64.4 \pm 1.2$ | $52.2 \pm 0.5$ | $76.8 \pm 0.6$ |
| | FINAL | $62.5 \pm 4.1$ | $55.6 \pm 2.0$ | $35.0 \pm 1.1$ | $32.0 \pm 2.0$ | $24.5 \pm 0.6$ | $19.1 \pm 0.5$ |
| | T-GAE (GNN$_c$) | $68.0 \pm 8.7$ | $82.0 \pm 3.4$ | $80.3 \pm 2.1$ | $78.6 \pm 1.9$ | $69.8 \pm 1.2$ | $72.3 \pm 2.4$ |
| | T-GAE (GCN) | $\mathbf{68.9 \pm 5.2}$ | $83.5 \pm 1.3$ | $82.2 \pm 1.4$ | $78.5 \pm 1.3$ | $62.3 \pm 1.6$ | $83.6 \pm 1.0$ |
| | T-GAE (GIN) | $49.6 \pm 16.2$ | $\mathbf{86.3 \pm 0.7}$ | $\mathbf{83.7 \pm 2.2}$ | $\mathbf{81.0 \pm 2.5}$ | $\mathbf{75.1 \pm 0.6}$ | $\mathbf{88.3 \pm 0.4}$ |
| **5% perturbation** | Spectral | $14.4 \pm 3.9$ | $28.8 \pm 9.4$ | $1.4 \pm 0.7$ | $3.5 \pm 1.9$ | $0.3 \pm 0.1$ | $0.5 \pm 0.2$ |
| | NetSimile | $12.6 \pm 1.4$ | $22.4 \pm 1.0$ | $10.3 \pm 0.4$ | $32.0 \pm 0.7$ | $18.4 \pm 0.1$ | $20.2 \pm 0.3$ |
| | FINAL | $\mathbf{24.3 \pm 2.3}$ | $20.5 \pm 1.6$ | $7.5 \pm 0.4$ | $8,6 \pm 0.7$ | $4.7 \pm 0.2$ | $3.5 \pm 1.3$ |
| | T-GAE (GNN$_c$) | $14.0 \pm 2.9$ | $28.9 \pm 2.8$ | $24.2 \pm 1.9$ | $38.5 \pm 2.3$ | $24.6 \pm 0.7$ | $10.8 \pm 0.4$ |
| | T-GAE (GCN) | $9.9 \pm 2.6$ | $28.3 \pm 1.3$ | $\mathbf{30.7 \pm 1.4}$ | $47.0 \pm 1.7$ | $22.2 \pm 0.7$ | $13.6 \pm 0.5$ |
| | T-GAE (GIN) | $13.0 \pm 1.8$ | $\mathbf{42.9 \pm 2.1}$ | $28.9 \pm 1.4$ | $\mathbf{52.4 \pm 3.3}$ | $\mathbf{35.7 \pm 0.8}$ | $\mathbf{23.6 \pm 0.6}$ |

Table 7: Graph Matching Accuracy for uniform perturbation model

and deleting edges, according to the degree Model perturbation probability presented in Section 5.3. Training and testing are conducted on different randomly generated perturbed samples. Table 8 reports the testing results of `T-GAE` when trained on the best perturbation levels.

| Algorithm | Celegans | Arenas | Douban | Cora | Dblp | Coauthor CS |
|---|---|---|---|---|---|---|
| **0% perturbation** | | | | | | |
| T-GAE | $89.7 \pm 1.4$ | $88.2 \pm 0.4$ | $\mathbf{90.2 \pm 0.4}$ | $87.6 \pm 0.4$ | $85.6 \pm 0.1$ | $97.6 \pm 0.1$ |
| T-GAE with pert. | $\mathbf{90.4 \pm 1.2}$ | $\mathbf{88.4 \pm 0.4}$ | $90.1 \pm 0.3$ | $\mathbf{87.6 \pm 0.4}$ | $\mathbf{85.7 \pm 0.1}$ | $\mathbf{97.6 \pm 0.0}$ |
| **1% perturbation** | | | | | | |
| T-GAE | $\mathbf{68.9 \pm 5.2}$ | $86.3 \pm 0.7$ | $83.7 \pm 2.2$ | $81.0 \pm 2.5$ | $\mathbf{75.1 \pm 0.6}$ | $88.3 \pm 0.4$ |
| T-GAE with pert. | $67.2 \pm 10.2$ | $\mathbf{87.3 \pm 1.0}$ | $\mathbf{85.3 \pm 1.0}$ | $\mathbf{82.7 \pm 0.8}$ | $74.1 \pm 0.7$ | $\mathbf{89.7 \pm 0.7}$ |
| **5% perturbation** | | | | | | |
| T-GAE | $14.0 \pm 2.9$ | $42.9 \pm 2.1$ | $30.7 \pm 1.4$ | $52.4 \pm 3.3$ | $35.7 \pm 0.8$ | $23.6 \pm 0.6$ |
| T-GAE with pert. | $\mathbf{17.7 \pm 2.0}$ | $\mathbf{55.7 \pm 3.2}$ | $\mathbf{47.1 \pm 2.2}$ | $\mathbf{57.6 \pm 1.1}$ | $\mathbf{39.4 \pm 0.7}$ | $\mathbf{30.2 \pm 0.7}$ |

Table 8: Performance Comparison of T-GAE when trained with/without perturbation on degree perturbation model

The results show that introducing training perturbations significantly improves the graph matching accuracy in high testing perturbation levels. When testing on low perturbation levels, however, the use of perturbations to train the `T-GAE` does not lead to significant changes in this perturbation model. At 5% testing perturbation, training with perturbations offers a 16.4% accuracy increase in the Douban dataset and 12.8% accuracy improvement in the Arenas Dataset, whereas at 1% testing perturbation, the increase is 1% and 1.6% respectively. And at 0 testing perturbation, the performance on these two datasets is similar.

## H.3 SPECIFIC PERTURBED TRAINING

The last experiment we conduct on the degree perturbation model is to compare the performance of `T-GAE` on Celegans, Arenas, Douban, and Cora, on 4 different settings: i) training is performed on a specific graph according to (8); ii) training on the family of all 4 graphs based on (9); iii) training on the perturbations of one specific graph according to (9), with $\mathbb{S}$ being defined over a family of graph perturbations; iv) training on the perturbations of all 4 graphs according to (10). The results for different levels of perturbations are presented in Table 6.

| Perturbation | Celegans | Arenas | Douban | Cora |
|---|---|---|---|---|
| | | 0% perturbation | | |
| T-GAE specific (8) | $88.9 \pm 1.1$ | $88.4 \pm 0.5$ | $90.0 \pm 0.3$ | $87.5 \pm 0.4$ |
| T-GAE (9) | $89.7 \pm 1.4$ | $88.2 \pm 0.4$ | $\mathbf{90.2 \pm 0.4}$ | $\mathbf{87.6 \pm 0.4}$ |
| T-GAE specific with pert. | $89.4 \pm 1.0$ | $88.3 \pm 0.4$ | $90.1 \pm 0.4$ | $87.5 \pm 0.6$ |
| T-GAE with pert. (10) | $\mathbf{90.4 \pm 1.2}$ | $\mathbf{88.4 \pm 0.4}$ | $90.1 \pm 0.3$ | $\mathbf{87.6 \pm 0.4}$ |
| | | 1% perturbation | | |
| T-GAE specific (8) | $\mathbf{74.8 \pm 2.8}$ | $85.8 \pm 1.5$ | $85.1 \pm 0.7$ | $82.3 \pm 1.3$ |
| T-GAE (9) | $68.9 \pm 5.2$ | $86.3 \pm 0.7$ | $83.7 \pm 2.2$ | $81.0 \pm 2.5$ |
| T-GAE specific with pert. | $70.7 \pm 4.6$ | $87.0 \pm 0.7$ | $84.1 \pm 0.8$ | $81.6 \pm 1.5$ |
| T-GAE with pert. (10) | $67.2 \pm 10.2$ | $\mathbf{87.3 \pm 1.0}$ | $\mathbf{85.3 \pm 1.0}$ | $\mathbf{82.7 \pm 0.8}$ |
| | | 5% perturbation | | |
| T-GAE specific (8) | $\mathbf{30.6 \pm 5.0}$ | $44.2 \pm 1.2$ | $41.3 \pm 1.6$ | $56.0 \pm 1.4$ |
| T-GAE (9) | $14.0 \pm 2.9$ | $42.9 \pm 2.1$ | $30.7 \pm 1.4$ | $52.4 \pm 3.3$ |
| T-GAE specific with pert. | $27.3 \pm 4.8$ | $46.4 \pm 3.8$ | $42.0 \pm 2.1$ | $56.3 \pm 1.8$ |
| T-GAE with pert. (10) | $17.7 \pm 2.0$ | $\mathbf{55.7 \pm 3.2}$ | $\mathbf{47.1 \pm 2.2}$ | $\mathbf{57.6 \pm 1.1}$ |

Table 9: Performance of T-GAE when trained on a family of graphs or on a specific graph on degree perturbation model

From Table 9 we observe that all the models work statistically the same when testing on zero perturbation. When testing on low or high perturbation levels, training with perturbation achieves the best accuracy in most datasets. The Celegans dataset is an exception and achieves a better matching accuracy by $5.9\%$ and $3.3\%$ when trained on a specific graph, respectively, compared to training on a family of graphs or perturbed adjacencies. Overall, we see that training with perturbations helps improve the matching accuracy and the robustness of the proposed method. `T-GAE with pert.` or `T-GAE specific with pert.`. In other words, training according to (10) is in general produces the most consistent and accurate results compared to the settings where we train the model on the original graph without perturbation.

From the additional perturbation model, we conclude that our proposed `T-GAE` framework is able to transfer and perform network alignment on large and small-size graphs and is robust to achieve consistent accuracy under different perturbation models and perturbation levels.

