# OpenReview forum: "Network Alignment with Transferable Graph Autoencoders"
_ICLR.cc/2024/Conference — Submitted to ICLR 2024_

### Official Review · Reviewer_yfvW · 2023-10-30

**Soundness:** 2 fair
**Presentation:** 2 fair
**Contribution:** 2 fair
**Rating:** 3
**Confidence:** 5

**Summary:**

The paper proposed T-GAE, a generalized graph autoencoder architecture which produces node embedding tailored to perform graph alignment which is equivalent to QAP. The authors drew the connection between GNNs and Spectral methods in graph matching. T-GAE uses transfer learning to perform network alignment on large graphs. Through extensive experimentation, the authors demonstrated the effectiveness of T-GAE compared to other baselines.

**Strengths:**

1. The authors established a connection between GNNs and spectral methods, and proved that there always exists a GNN that can perform at least as well as the spectral approach. The authors provided comprehensive proofs for these theorems.

2. The authors proposed a self-supervised framework, that can also scale for large graphs by leveraging transfer learning

3. T-GAE empirically outperformed the baselines on graph matching and subgraph matching tasks, especially at higher perturbation levels.

**Weaknesses:**

1. The authors did not compare their model with recent state-of-the-art models like S-GWL[1], Cone-Align[2] which could have been good baselines to compare against.
2. The runtimes are not provided for Graph matching datasets. Fig. 5 in appendix has runtime comparison only for Subgraph matching tasks.
3. The authors mentioned that T-GAE takes more time to train on larger networks compared to WAlign but did not provide specific training times for GNN-based methods, which would offer a better context for this statement.
4. It appears in section 5.3, Fig. 3 should be referenced but instead Fig. 5 is referenced
5. README.txt in the repository mentions that the submission is for Neurips 2023.

References:
* [1] Hongteng Xu, Dixin Luo, and Lawrence Carin. 2019. Scalable GromovWasserstein learning for graph partitioning and matching. NeurIPS 32 (2019)
* [2] Xiyuan Chen, Mark Heimann, Fatemeh Vahedian, and Danai Koutra. 2020. CONE-Align: Consistent Network Alignment with Proximity-Preserving Node Embedding. In CIKM. ACM, 1985–1988.

**Questions:**

1. Why did you not compare with state-of-the-art recent approaches like S-GWL[1] and Cone-Align[2]?
2. What are the runtimes for Graph matching tasks? Runtime comparison is provided only for Subgraph matching tasks. Also, what are the training times for GNN-based methods?
3. Why are the accuracy scores of the degree-perturbation model less than the uniform-probability perturbation model, at higher perturbation level?

---

> ### Author Response · Authors · 2023-11-17
>
> > The authors did not compare their model with recent state-of-the-art models like S-GWL, Cone-Align which could have been good baselines to compare against.
>
>
> We thank the reviewer for their comment. We add the requested experiments below, along with the results of the proposed T-GAE. Note that for T-GAE, we report the best accuracy among all 3 backbones we experimented with.
>
> ```markdown
> |-----------------------------------------------------------------------|
> |ConeAlign for Graph Matching                                           |
> |-----------------------------------------------------------------------|
> |Pert| Celegans |   Arena  |  Douban  |   Cora   |   DBLP   |Coauthor_CS|
> |----|----------|----------|----------|----------|----------|-----------|
> | 0  | 66.6+-1.2| 75.6+-0.4| 68.1+-0.4| 38.5+-0.7| 44.3+-0.6| 75.8+-0.5 |
> |----|----------|----------|----------|----------|----------|-----------|
> | 1% | 60.9+-2.5| 73.2+-0.8| 64.7+-0.4| 33.5+-1.6| 37.8+-1.1| 68.2+-2.8 |
> |----|----------|----------|----------|----------|----------|-----------|
> | 5% | 50.5+-3.4| 65.1+-1.1| 54.1+-1.2| 23.0+-2.0| 24.4+-2.9| 51.4+-5.1 |
> |----|----------|----------|----------|----------|----------|-----------|
> |S-GWL for Graph Matching                                               |
> |-----------------------------------------------------------------------|
> |Pert| Celegans |   Arena  |  Douban  |   Cora   |   DBLP   |Coauthor_CS|
> |----|----------|----------|----------|----------|----------|-----------|
> | 0  | 93.0+-1.5| 0.1+-0.1 | 90.1+-0.3| 87.3+-0.7|> 10 hours|> 10 hours |
> |----|----------|----------|----------|----------|----------|-----------|
> | 1% | 92.4+-1.4| 0.1+-0.1 | 0.0+-0.0 | 8.7+-26.0|> 10 hours|> 10 hours |
> |----|----------|----------|----------|----------|----------|-----------|
> | 5% |81.1+-27.0| 0.1+-0.1 | 0.0+-0.0 |16.3+-32.6|> 10 hours|> 10 hours |
> |----|----------|----------|----------|----------|----------|-----------|
> |T-GAE for Graph Matching                                               |
> |-----------------------------------------------------------------------|
> |Pert| Celegans |   Arena  |  Douban  |   Cora   |   DBLP   |Coauthor_CS|
> |----|----------|----------|----------|----------|----------|-----------|
> | 0  |89.5+-1.3 |88.4+-0.5 |90.1+-0.4 |87.4+-0.4 |85.8+-0.1 |97.6+-0.1  |
> |----|----------|----------|----------|----------|----------|-----------|
> | 1% |84.1+-1.1 |84.8+-0.6 |84.9+-0.6 |82.9+-0.5 |79.1+-0.4 |86.5+-0.8  |
> |----|----------|----------|----------|----------|----------|-----------|
> | 5% |50.8+-3.3 |47.1+-5.6 |57.9+-6.1 |58.2+-2.0 |40.8+-2.1 |26.9+-5.4  |
> |----|----------|----------|----------|----------|----------|-----------|
>
> ```
>
> Our observations are: (1) The performance of ConeAlign does not drop dramatically as we add perturbations to the testing data. but its performance has been consistently worse than the proposed T-GAE. (2) The performance of S-GWL is very unstable, it achieves state-of-the-art performance on Celegans, but got inferior matching accuracy on Arena. For Douban and Cora, it matches well on the original graphs, but the matching accuracy vastly drops when we add perturbations. (3) The method is not scalable for its running time.
>
> Note that we only test these two methods on graph matching tasks, as they are designed to match graphs with same number of nodes.

---

> > ### Comment · Reviewer_yfvW · 2023-11-19
> > **What is the source of major discrepancy between the presented results and those reported in the literature?**
> >
> > First of all, I thank the authors for their efforts in addressing the queries.
> >
> > With respect to the new results, I am concerned over the reported accuracy values of S-GWL. In Fig. 1 of the paper [1], the accuracy values of S-GWL on the Arenas dataset are documented, demonstrating consistent performance with an accuracy exceeding 80% up to 5% noise. However, the accuracy values presented in your response are nearly 0, marking a substantial discrepancy. Additionally, the instability attributed to S-GWL in your response is not evident in the findings reported in the aforementioned paper [1].
> >
> > I am interested in understanding the reasons behind the notable deviation between the accuracy values provided in your response and those documented in the existing literature.
> >
> > References:
> > [1] CSkitsas, K., Orlowski, K., Hermanns, J., Mottin, D., & Karras, P. (2023). Comprehensive Evaluation of Algorithms for Unrestricted Graph Alignment. In J. Stoyanovich, J. Teubner, N. Mamoulis, E. Pitoura, & J. Mühlig (Eds.), Proceedings 26th International Conference on Extending Database Technology, EDBT 2023, Ioannina, Greece, March 28-31, 2023 (pp. 260–272). OpenProceedings.org. https://doi.org/10.48786/edbt.2023.21

---

> > > ### Author Response · Authors · 2023-11-20
> > >
> > > We appreciate the reviewer for raising this fair question. The answer follows:
> > >
> > > (1) The two Arena graphs used in our experiments and in the survey paper are not totally the same. The dataset we are using is more challenging for the competing baselines to handle as it contains isolated nodes. To see this, we give a plot of the Laplacian eigenvalues of the dataset that is used in the survey (Skitsas et al.,2023) and that of the dataset used in our experiments (all plots are put on the Github repository https://github.com/GraphMatching/Graph-Matching). From our analysis and previous analyses we observe that S-GWL is very sensitive when it comes to matching graphs with isolated nodes and that is why it completely fails in the initial Arena experiment. Next we conduct experiments on our tested Arena dataset after removing the isolated nodes and on the dataset that is used in the survey paper.
> > >
> > > ```markdown
> > > |------------------------------------------|
> > > |Graph Matching accuracy on Arena dataset  |
> > > |used in our experiment after removing     |
> > > |isolated nodes                            |
> > > |---------|----------|----------|----------|
> > > |         |  0 pert  |  1% pert |  5% pert |
> > > |---------|----------|----------|----------|
> > > |   TGAE  |95.0+-0.4 |91.6+-0.5 |61.7+-5.3 |
> > > |---------|----------|----------|----------|
> > > |  S-GWL  |94.6+-0.5 |28.7+-43.7|37.4+-45.8|
> > > |---------|----------|----------|----------|
> > > |Graph Matching accuracy on Arena dataset  |
> > > |used in the survey paper.                 |
> > > |---------|----------|----------|----------|
> > > |         |  0 pert  |  1% pert |  5% pert |
> > > |---------|----------|----------|----------|
> > > |   TGAE  |97.5+-0.4 |94.0+-0.7 |58.9+-6.6 |
> > > |---------|----------|----------|----------|
> > > |  S-GWL  |97.5+-0.3 |87.9+-29.3| 0.1+-0.1 |
> > > |---------|----------|----------|----------|
> > >
> > > ```
> > > (2) In our experiments the perturbation model is also different compared to the one used in the review paper. More importantly, for every setting we reported in the manuscript, we run the experiment on 10 random perturbations of the graph and report the mean accuracy and standard deviation, whereas in the survey paper only one single experiment is reported for every noise level. Although it is clear that S-GWL is sensitive to graphs with isolated nodes, these nodes are not guaranteed to be removed by our perturbation model.
> > > Since the graph used in the review paper has many nodes with degree of 1 (please refer to the node degree distribution graphs on Github), there is higher chance that these nodes could become isolated nodes after perturbation. This explains the performance drop of S-GWL when noise is added.
> > >
> > > (3) We append the different versions of Arenas data in our Github repository(https://github.com/GraphMatching/Graph-Matching) for the reviewer to check. The Arena dataset used in our experiment after removing isolated edges is contained in the file "arena_no_iso.pt" and the dataset used in the survey paper is contained in "arena_dense.pt". For S-GWL, we are using the official implementation which can be found in https://github.com/HongtengXu/s-gwl.

---

> > > > ### Author Response · Authors · 2023-11-22
> > > > **Discrepancy clarified. We would like to hear your thoughts.**
> > > >
> > > > Dear reviewer,
> > > >
> > > > Thank you for your input which has been pivotal in improving our paper. We believe we have addressed all of your concerns including your comment about the performance of S-GWL. We would be happy if you could engage in our fruitful discussion and give us your thoughts on the revised manuscript and our response

---

> > > > > ### Comment · Reviewer_yfvW · 2023-11-22
> > > > > **thanks for the clarifications, but it requires better contextualization**
> > > > >
> > > > > I appreciate the authors for conducting additional experiments and shedding light on the discrepancy between reported results in the literature and those presented in the revision. However, the new results further emphasize that the technique tends to outperform existing state-of-the-art (SOTA) algorithms like S-GWL only on specific types of datasets. On regular benchmark datasets for network alignment, the improvement is relatively minor over S-GWL (which is considered the state of the art and was initially omitted).
> > > > >
> > > > > Therefore, while the overall work shows promise, it necessitates significant modifications to present a more comprehensive picture. Specifically, the following points should be addressed:
> > > > >
> > > > > 1. Clearly motivate the use of special datasets (such as datasets with isolated nodes) where the improvement over S-GWL is substantial. Explain why these datasets are relevant for real-world workloads, as the current introduction is too generic.
> > > > >
> > > > > 2. Explore whether isolated nodes are the sole reason for S-GWL's failure. Are there other topological properties that contribute to this?
> > > > >
> > > > > 3. The current draft does not incorporate the discussion from the previous response, which better elucidates why results from the reported literature are inconsistent. This information is currently mentioned only in the open review comments.
> > > > >
> > > > > Overall, the manuscript requires substantial repositioning in light of the findings made during the discussion.

---

> > > > > > ### Author Response · Authors · 2023-11-22
> > > > > > **Thank you for your engagement**
> > > > > >
> > > > > > We thank the reviewer for their continued engagement. Our interactions have been fruitful to us and we hope that they have been valuable to you as well.
> > > > > >
> > > > > > We want to point out that at this point the reviewer and us are roughly in agreement that the method we propose is valuable. In some cases, we attain comparable performance to S-GWL and in some other cases, we attain significantly better performance, especially when the graphs are sparse or large. We never do much worse.
> > > > > >
> > > > > > We understand that the reviewer thinks that the experiments in which we do better are not significant, but with all due respect, we disagree. We are using standard datasets in network alignment (Arenas, ACM/DBLP, Douban Online/Offline) and some larger ones (Dblp, Coauthor CS) that test scalability. In any case, we hope that we can agree that our work is still something that has value for the community.
> > > > > >
> > > > > > The reviewer is right that we should incorporate the discussion from the previous response, and we indeed included it at the end of page 7 and the beginning of page 8. We are prioritizing engagement at the moment, which as we said above, has been fruitful and enjoyable.

---

> ### Author Response · Authors · 2023-11-17
>
> >The runtimes are not provided for Graph matching datasets. Fig. 5 in appendix has runtime comparison only for Subgraph matching tasks.
>
> We thank the reviewer for their suggestion. Below we measure the runtimes of the competing approaches. The results reported are tested on matching graphs without perturbations. For each dataset, we report the embedding time and matching time on 2 graphs at 0 perturbation level.
>
> ```markdown
> Runtime (embedding inference+matching) in second for Graph Matching
> |         | Celegans |   Arena  |  Douban  |   Cora   |   DBLP   |Coauthor_CS|
> |---------|----------|----------|----------|----------|----------|-----------|
> |   TGAE  |  0.098   |  0.241   |   1.934  |   0.842  |  173.836 |  193.289  |
> |---------|----------|----------|----------|----------|----------|-----------|
> |   GAE   |  0.094   |  0.252   |   1.949  |   0.857  |  174.410 |  194.062  |
> |---------|----------|----------|----------|----------|----------|-----------|
> |  WAlign |  0.0784  |  0.205   |   1.800  |   0.766  |  169.694 |  189.032  |
> |---------|----------|----------|----------|----------|----------|-----------|
> |Spectrul |  5.712   |  2.819   |  60.298  |  54.770  | > 10hours| > 10hours |
> |---------|----------|----------|----------|----------|----------|-----------|
> |GraphWave|  8.308   | 32.994   | 281.629  | 131.230  | 5470.724 |  6291.368 |
> |---------|----------|----------|----------|----------|----------|-----------|
> |ConeAlign|  1.333   |  3.500   |  31.955  |  13.799  |  887.090 |  1099.145 |
> |---------|----------|----------|----------|----------|----------|-----------|
> |  S-GWL  | 27.844   | 37.443   |3394.522  | 311.201  | > 10hours| > 10hours |
> |---------|----------|----------|----------|----------|----------|-----------|
> |  FINAL  |  0.030   |  0.081   |   1.007  |   0.498  |  86.788  |   118.065 |
> |---------|----------|----------|----------|----------|----------|-----------|
> ```
>
>
> >The authors mentioned that T-GAE takes more time to train on larger networks compared to WAlign but did not provide specific training times for GNN-based methods, which would offer a better context for this statement.
>
> > What are the training times for GNN-based methods?
>
> Thank you for the suggestion. Figure 5, in the original manuscript, reports the time each baseline needs to generate the node embeddings used for matching. For GNN methods this time corresponds to training time.  We will clarify that in the revised manuscript.
>
>
> > It appears in section 5.3, Fig. 3 should be referenced but instead Fig. 5 is referenced
>
> Thanks for pointing this out, we will fix in the revised paper.
>
> > README.txt in the repository mentions that the submission is for Neurips 2023.
>
> It has been modified from the previous submission.
>
>
> >Why are the accuracy scores of the degree-perturbation model less than the uniform-probability perturbation model, at higher perturbation level?
>
> The uniform model is agnostic to the graph structure. On the other hand, the degree perturbation model adds or removes edges according to the degree of each node. For example, edges between nodes with high degree are more likely to be removed and edges are more likely to be added between nodes with high degree. As a result, the degree perturbation model changes the graph structure more drastically compared to the uniform perturbation model, and the task of alignment at high perturbation levels becomes more challenging.

---

### Official Review · Reviewer_zEkT · 2023-10-31

**Soundness:** 3 good
**Presentation:** 2 fair
**Contribution:** 3 good
**Rating:** 6
**Confidence:** 4

**Summary:**

This paper proposes a network alignment technique using GNN, which is called T-GAE. Compared with other network alignment approaches, T-GAE is capable of transferring to other unseen graphs, and as such, the alignment can also be performed without retraining. Specifically, T-GAE devises a GNN encoder to match nodes from different graphs. Through permutation and perturbation, different versions of graphs are generated to be augmented datasets. These datasets enable T-GAE to have generalization and be extended to other graphs. The transferability helps T-GAE perform network alignment on other graphs without retraining.

**Strengths:**

+ The generated embeddings are theoretically more capable compared with spectral methods. The theorem provides a lower bound that supports T-GAE to have high performance in network alignment.
+ According to the experiments, T-GAE performs well in transferability, which avoids plenty of retraining. Moreover, T-GAE can be extended to graphs with larger sizes, which is much more scalable compared with existing methods.
+ T-GAE is robust with regard to the permutation noise.

**Weaknesses:**

- There are many typos, e.g., Figure 1’s caption: traing -> training; Figure 5: Spectrul -> Spectral; Theorem 3.2: a solutions -> solutions. What does 1 mean in the 5th line in Appendix H.1?
- ⋆ and * are too similar in Theorem and it is hard to distinguish them. I suggest the authors could substitute them with some other symbols.
- Theorem 3.2 looks sound, but not all the adjacencies have non-repeated eigenvalues, so the lower bound is not always satisfied.
- Some other concerns can be referred to in the `Questions’ section.

**Questions:**

- How is S generated from a family of graphs? In the experiments, does S mean the adjacency matrices and their permuted-perturbed versions? How can this family of graphs be extended to other untrained graphs? I understand that the permutation and perturbation can augment the datasets but they still cannot cover the distribution of the unseen datasets.
- Are the 10 randomly perturbed samples the same in training and test on Celegans, Arena, Douban, and Cora networks?
- Does perturbation widely exist in real-world datasets?
- I notice that the perturbation is also reported in FINAL, and they even test the 20% perturbation noise while 5% is reported for T-GAE. However, the performance of FINAL is very stable but is quite bad in Table 3. Please explain the reason.
- Why are the results of graphwave not reported on DBLP and Coauthor CS?
- What is the future work for T-GAE?

---

> ### Author Response · Authors · 2023-11-17
>
> > There are many typos, e.g., Figure 1’s caption: traing -> training; Figure 5: Spectrul -> Spectral; Theorem 3.2: a solutions -> solutions. What does 1 mean in the 5th line in Appendix H.1?
>
> We thank the reviewer for pointing out these typos, we will modify them in the revised version of the paper. 1 in the 5th line in Appendix H.1 is a typo. We are currently working on the revised version and will upload later.
>
> > ⋆ and * are too similar in Theorem and it is hard to distinguish them. I suggest the authors could substitute them with some other symbols.
>
> We appreciate this suggestion to improve our presentation of the theorem, and we will replace it with a more readable symbol in the revised paper.
>
> >Theorem 3.2 looks sound, but not all the adjacencies have non-repeated eigenvalues, so the lower bound is not always satisfied.
>
> The reviewer is right that there exist different graphs with the same set of eigenvalues which are not considered in our theorem. However, this class of graphs is small [1] (it involves certain tree structures) and the vast majority real-world graphs have different eigenvalues. Our experiments also support Theorem 3.2, as T-GAE outperforms the spectral method. We add this discussion in the revised manuscript.
>
> >How is S generated from a family of graphs? In the experiments, does S mean the adjacency matrices and their permuted-perturbed versions? How can this family of graphs be extended to other untrained graphs? I understand that the permutation and perturbation can augment the datasets but they still cannot cover the distribution of the unseen datasets.
>
>
> In equation (10) of Section 4.3, S is the family of graphs that we train the GNN on (in case of Table 3, it consists of Celegans, Arena, Douban, Cora). M is the family of graph perturbations, i.e., the augmented graphs. We cannot cover the distribution of all unseen graphs but our approach is empirically proven to be effectively transferable to the out-of-distribution large-scale networks (DBLP and Coauthor_CS in Table 3). We believe this result to be significant, considering the different types of networks involved in the experiments. We train with graphs from different domains (distributions), i.e., citation, social, email communication, and interactome networks. The transferability of Graph Neural Networks (Ruiz et al,.2020) suggests that we can train the GNN on smaller graphs and execute efficiently on larger scale graphs when the substructures (motifs) in the tested networks are partially observed during training. Thus, we propose in equation (9) and (10) to train the GNN on multiple graphs to have more kinds of motifs observed during training, which is not achievable by a classical graph autoencoder. Please also refer to Remark 4.1 for this discussion.

---

> ### Author Response · Authors · 2023-11-17
>
> >Are the 10 randomly perturbed samples the same in training and test on Celegans, Arena, Douban, and Cora networks?
>
> The graphs used in training are different from those used for testing. For Table 3 we train on the original adjacency matrix and test on the perturbed graphs. For Table 4 we train on 10 randomly generated perturbations and test on another set of 10 randomly perturbed graphs. The random perturbation generation process is described in Section 5.2 and Appendix H.
>
> >Does perturbation widely exist in real-world datasets?
> I notice that the perturbation is also reported in FINAL, and they even test the 20% perturbation noise while 5% is reported for T-GAE. However, the performance of FINAL is very stable but is quite bad in Table 3. Please explain the reason.
>
> Thank you for pointing this out. We use a different perturbation model from FINAL. Specifically, we change the graph structure by adding or removing edges, whereas in FINAL, the edges remain the same, but the weight on each edge is changed by adding some noise. It is therefore clear that our perturbation model is more practical and more challenging for the baselines to handle.
>
>
> >Why are the results of graphwave not reported on DBLP and Coauthor CS?
>
> GraphWave is not reported on DBLP and Coauthor CS because the implementation breaks, i.e., it is not scalable to large-scale graphs.
>
> >What is the future work for T-GAE?
>
>
>
> We appreciate the reviewer for motivating us to think about the future work. Our plan is to extend the TGAE work in three directions. First, we intend to design more robust node embeddings using ideas from constrained and resilient learning. Second, we want to further improve the scalability of our approach to graphs with million nodes. Finally, we plan to use our method as part of other architecures that are task specific. An area that is interesting to us is circuit placement.
>
>
> [1] Haemers, W.H. and Spence, E. Enumeration of cospectral graphs.European Journal of Combinatorics, 25(2):199211, 2004.

---

> > ### Comment · Reviewer_zEkT · 2023-11-18
> > **Follow-up to author response**
> >
> > I appreciate the authors' effort and clear clarifications. I believe that the authors have addressed most of my questions:
> >
> > 1) Theorem 3.2 cannot cover all the circumstances, but the failed case is limited.
> > 2) The generality is supported by the empirical results.
> > 3) There exists the inconsistence of perturbation noise between this paper and the paper 'FINAL', but I agree with the author's clarification of the different definitions of 'perturbation noise' and especially, the definition in this paper is stricter.
> >
> > Based on the above reasons, I raise the score from 5 to 6. I still expect the revised manuscript and I suggest the author could emphasize 1) and illustrate 3) in the updates. I will keep following this paper and discussing with other reviewers to make a final decision.

---

> > > ### Author Response · Authors · 2023-11-21
> > >
> > > We thank the reviewer for appreciating our work and raising their score. We have included discussions of (1) when presenting the theorem and provided more details and empirical evidence of our experiments in the revised manuscript.

---

### Official Review · Reviewer_vYiA · 2023-10-31

**Soundness:** 3 good
**Presentation:** 4 excellent
**Contribution:** 3 good
**Rating:** 8
**Confidence:** 4

**Summary:**

In the submitted manuscript, the authors propose a graph autoencoder model (T-GAE) to perform the task of network alignment. They prove that alignment results obtained from their T-GAE are at least as accurate as traditional alignment methods based on the elementwise absolute value of adjacency matrix eigenvectors. They furthermore propose to augment the training dataset by perturbing and permuting training graphs to yield a GAE, which is more robust to noise and potentially more transferable between datasets. In a range of experiments they find their T-GAE model to outperform several baseline models on the task of network alignment.

**Strengths:**

- The ideas are clearly presented and it is very easy to follow your writing.
- The complexity analysis in Section 4.4 and the Limitations paragraph in Section 5.3 are nice additions.
- The approach you propose is well-motivated (so well in fact that I am surprised that it has not been published already).

**Weaknesses:**

- While the general ideas are clearly presented I would have liked to see more technical detail in several places. Generally, your presentation of the concepts sometimes was a little superficial (for example would it be nice if you extended Section 4 to also describe your proposed model in greater detail).
- Several of the statements made in your paper are either too general to be true (see Question 1).
- The experimental evaluation could likely be further improved (see Question 3).

**Questions:**

1] It seems to me that several of the claims you make in the paper are stated too generally to be true and cannot be substantiated by evidence, e.g., "enable a generalized framework that can be applied to out-of-distribution graphs that have not been observed during training" (there is no guarantee that you can perform well on out-of-distribution graphs, in fact we already observe strong performance drops at 5% perturbation rates; this should be clarified) and "the learned mapping can be applied to larger, big-data graphs with high accuracy and efficiency", as well as "to tackle network alignment at a very large scale" (the largest network you consider has 18 thousand nodes and 81 thousand edges, in the context of the ogb benchmarks [1] for example it is difficult to call your considered data sets "very large" or even "big-data"). I therefore want to ask you to please weaken these statements to be more realistic or to provide further evidence to substantiate these claims in the generality in which they are stated.

2] In Section 4.3 in your data augmentation method you propose to permute the node labels of the perturbed graphs. GNNs should either be permutation equivariant or invariant to node permutation, in either case the permutation of node labels in your perturbed graphs should be inconsequential. Could you therefore please motivate the node permutation in your data augmentation method?

3] Your experimental evaluation could be further improved.

3.1] Using a standard graph autoencoder and graph variational autoencoder as a baseline seems intuitive to me and would provide a nice ablation study to give insight on which parts of your architecture are major contributors to your observed performance increases.

3.2] While you state DeepWalk to be one of your baseline models, it is absent in Table 3 and only used for subgraph matching in Figure 3. Could you please explain why you choose not to use it for graph matching?

3.3] While this is certainly not a requirement, it might be worthwhile to include more advanced unsupervised embedding methods such as LINE [2] in your baselines to strengthen the evidence in favour of your model.

3.4] The larger datasets you transfer to are known to be structurally similar to the smaller datasets you trained on, e.g., they are also homophilic. It might be more meaningful to consider to what degree your pre-trained models transfer to datasets with drastically different characteristica, such as for example heterophilic datasets. Or whether your method is still viable at even larger scale than 18 thousand nodes.


4] Minor comments:

4.1] The font in Figure 1 is too small to be comfortably read on a print-out.

4.2] It is unclear to me what you mean by a GNNs output being unique. It would be good if you could clarify your use of the word unique in the paper. GNNs can produce equal representations of different graphs (see for example [3]) and consequently, one could reasonably claim GNN embeddings to not be unique in the sense that GNN embeddings do not uniquely identify graphs they correspond to.

4.3] There is minor errors in the bold font setting in Tables 3 and 4, e.g., "90.0\pm0.4" in row 1 column 3 of Table 3 and "90.1\pm0.4" in row 1 column 3 Table 4.

4.4] I could not see where you describe or reference the GNN_c, which seems to be one of the backbones of your T-GAE. Could you please add further detail on this model or point me to the part of your paper where this is described?


[1] Hu, W., Fey, M., Zitnik, M., Dong, Y., Ren, H., Liu, B., Catasta, M. and Leskovec, J., Open graph benchmark: Datasets for machine learning on graphs. Advances in neural information processing systems, 33, pp.22118-22133. 2020.

[2] Tang, J., Qu, M., Wang, M., Zhang, M., Yan, J. and Mei, Q., May. Line: Large-scale information network embedding. In Proceedings of the 24th international conference on world wide web (pp. 1067-1077). 2015.

[3] Xu, K., Hu, W., Leskovec, J. and Jegelka, S. How powerful are graph neural networks?. ICLR. 2019.

---

> ### Author Response · Authors · 2023-11-17
>
> > While the general ideas are clearly presented I would have liked to see more technical detail in several places. Generally, your presentation of the concepts sometimes was a little superficial (for example would it be nice if you extended Section 4 to also describe your proposed model in greater detail).
>
> Thank you for the kind suggestion on the organization of our paper. We have included the model details in Section 5.2 of the revised manuscript.
>
>
> > It seems to me that several of the claims you make in the paper are stated too generally to be true and cannot be substantiated by evidence, e.g., "enable a generalized framework that can be applied to out-of-distribution graphs that have not been observed during training" (there is no guarantee that you can perform well on out-of-distribution graphs, in fact we already observe strong performance drops at 5% perturbation rates; this should be clarified) and "the learned mapping can be applied to larger, big-data graphs with high accuracy and efficiency", as well as "to tackle network alignment at a very large scale" (the largest network you consider has 18 thousand nodes and 81 thousand edges, in the context of the OGB benchmarks for example it is difficult to call your considered data sets "very large" or even "big-data"). I therefore want to ask you to please weaken these statements to be more realistic or to provide further evidence to substantiate these claims in the generality in which they are stated.
>
> We agree with the reviewer that some of the statements should be modified to be more precise. However, there is both empirical and theoretical evidence that we can transfer well to unseen graphs. In the experimental front we see that our method achieves very high accuracy in DBLP and Coauthor_CS, although they were not observed during training. Furthermore transferability results on GNNs (Ruiz et al., 2020), theoretically motivate and guarantee (to a certain extend) the success of our work.
>
> Regarding your comment on the performance drop at 5%, note that we are implementing a challenging perturbation model (we change of graph structure). It is therefore very likely that this performance drop is not due to the potential limitations of our approach, but could be attributed to the fact that the alignment task becomes unrealistic beyond 5% perturbation.
>
> The reviewer is correct that for tasks such as node classification or link prediction, the datasets we used for experiments could not be described as "very large scale". Following your comment we are changing "very lagre-scale" statements to "large-scale" in the revised version. Note however that in the context of graph matching we are conducting the largest scale experiments to the best of our knowledge. For example, the largest graph used in WAlign has 3.9k nodes, and the largest graph used for ConeAlign and S-GWL (new baselines to be included in the paper) are of 4k nodes. We have included these discussions in the revised manuscript to avoid confusions.
>
> > In Section 4.3 in your data augmentation method you propose to permute the node labels of the perturbed graphs. GNNs should either be permutation equivariant or invariant to node permutation, in either case the permutation of node labels in your perturbed graphs should be inconsequential. Could you therefore please motivate the node permutation in your data augmentation method?
>
> The reviewer is right that GNNs are permutation equivariant and there is no need to add the permutation step in our approach. However, since the perturbation/permutation model is also used to assess the performance of the competing algorithms (that do not necessarily exhibit permutation equivariance), node permutation is used as a safety to guarantee fair comparisons.

---

> ### Author Response · Authors · 2023-11-17
>
> > Using a standard graph autoencoder and graph variational autoencoder as a baseline seems intuitive to me and would provide a nice ablation study to give insight on which parts of your architecture are major contributors to your observed performance increases.
>
> We agree with the reviewer that using vanilla GAE and VGAE as baselines could help to back up the main claims of this paper. We have run both of them and present the result in the following table. We also append the results for the proposed T-GAE framework. Note that for T-GAE we report the best accuracy from the 3 backbone message passing mechanisms.
>
> ```markdown
> |-----------------------------------------------------------------------|
> |GAE for Graph Matching                                                 |
> |-----------------------------------------------------------------------|
> |Pert| Celegans |   Arena  |  Douban  |   Cora   |   DBLP   |Coauthor_CS|
> |----|----------|----------|----------|----------|----------|-----------|
> | 0  |80.2+-2.6 |76.4+-1.1 |65.6+-0.4 |84.0+-0.8 |74.1+-0.3 |71.1+-0.3  |
> |----|----------|----------|----------|----------|----------|-----------|
> | 1% |13.6+-11.9| 9.0+-6.2 | 3.3+-2.9 |12.5+-5.1 | 3.9+-0.9 | 3.0+-1.2  |
> |----|----------|----------|----------|----------|----------|-----------|
> | 5% | 3.9+-5.1 | 1.3+-0.8 | 0.6+-0.6 | 2.2+-0.9 | 0.5+-0.1 | 0.1+-0.0  |
> |----|----------|----------|----------|----------|----------|-----------|
> |T-GAE for Graph Matching                                               |
> |-----------------------------------------------------------------------|
> |Pert| Celegans |   Arena  |  Douban  |   Cora   |   DBLP   |Coauthor_CS|
> |----|----------|----------|----------|----------|----------|-----------|
> | 0  |89.5+-1.3 |88.4+-0.5 |90.1+-0.4 |87.4+-0.4 |85.8+-0.1 |97.6+-0.1  |
> |----|----------|----------|----------|----------|----------|-----------|
> | 1% |84.1+-1.1 |84.8+-0.6 |84.9+-0.6 |82.9+-0.5 |79.1+-0.4 |86.5+-0.8  |
> |----|----------|----------|----------|----------|----------|-----------|
> | 5% |50.8+-3.3 |47.1+-5.6 |57.9+-6.1 |58.2+-2.0 |40.8+-2.1 |26.9+-5.4  |
> |----|----------|----------|----------|----------|----------|-----------|
>
>
> |--------------------------------------------|
> |GAE for Sub-graph Matching                  |
> |--------------------------------------------|
> |Hit Ratio| ACM/DBLP | Douban Online/Offline |
> |---------|----------|-----------------------|
> |  Hit@1  |   0.83   |        0.54           |
> |---------|----------|-----------------------|
> |  Hit@5  |   3.28   |        1.34           |
> |---------|----------|-----------------------|
> |  Hit@10 |   5.43   |        2.95           |
> |---------|----------|-----------------------|
> |  Hit@50 |  10.51   |       13.95           |
> |---------|----------|-----------------------|
> |  Hit@100|  14.09   |       23.97           |
> |---------|----------|-----------------------|
> |T-GAE for Sub-graph Matching                |
> |--------------------------------------------|
> |Hit Ratio| ACM/DBLP | Douban Online/Offline |
> |---------|----------|-----------------------|
> |  Hit@1  |   68.95  |        26.39          |
> |---------|----------|-----------------------|
> |  Hit@5  |   88.60  |        45.97          |
> |---------|----------|-----------------------|
> |  Hit@10 |   92.96  |        56.17          |
> |---------|----------|-----------------------|
> |  Hit@50 |   97.21  |        83.18          |
> |---------|----------|-----------------------|
> |  Hit@100|   98.14  |        91.60          |
> |---------|----------|-----------------------|
> ```
> We observe that the matching accuracy of vanilla GAE drops quickly with the increase of perturbation levels on graph matching tasks. This implies that GAE cannot effectively handle the distribution shift between the two graphs brought by the structural dissimilarity. This is further supported by the results on sub-graph matching tasks, where GAE consistently generates poor accuracy when trying to match two different real graphs.
>
> We also tested the performance of VGAE on the 4 smaller datasets and it could not produce meaningful matching. The result is expected as VGAE is not permutation equivariant because of the Gaussian noise step.
>
> ```markdown
> VGAE for Graph Matching
> |Pert|  Celegans  |    Arena    |    Douban  |    Cora    |
> |----|------------|-------------|------------|------------|
> | 0  |  0.3+-0.1  |   0.1+-0.1  |  0.0+-0.0  |  0.1+-0.0  |
> |----|------------|-------------|------------|------------|
> | 1% |  0.3+-0.1  |   0.1+-0.1  |  0.0+-0.0  |  0.1+-0.1  |
> |----|------------|-------------|------------|------------|
> | 5% |  0.6+-0.3  |   0.2+-0.1  |  0.0+-0.0  |  0.1+-0.0  |
> |----|------------|-------------|------------|------------|
>
> ```
>
>
> The previous results and discussions are included in the revised manuscript.

---

> ### Author Response · Authors · 2023-11-17
>
> > While you state DeepWalk to be one of your baseline models, it is absent in Table 3 and only used for subgraph matching in Figure 3. Could you please explain why you choose not to use it for graph matching?
>
> The reason why we omit DeepWalk from graph matching tasks is because it produces inferior performance as shown in the following table. This performance can be attributed to the fact that Deepwalk is not permutation equivariant as it can produce different embeddings for the same graph in different runs. We will clarify this in the revised manuscript.
>
> ```markdown
> DeepWalk for Graph Matching
> |Pert|  Celegans  |    Arena    |    Douban  |    Cora    |
> |----|------------|-------------|------------|------------|
> | 0  |  1.8+-0.6  |  0.3+-0.2   |  0.1+-0.0  |  0.1+-0.0  |
> |----|------------|-------------|------------|------------|
> | 1% |  1.2+-0.5  |  0.3+-0.1   |  0.1+-0.0  |  0.2+-0.1  |
> |----|------------|-------------|------------|------------|
> | 5% |  1.0+-0.3  |  0.2+-0.1   |  0.0+-0.0  |  0.1+-0.0  |
> |----|------------|-------------|------------|------------|
> ```
>
> > While this is certainly not a requirement, it might be worthwhile to include more advanced unsupervised embedding methods such as LINE in your baselines to strengthen the evidence in favour of your model.
>
>
> We appreciate the reviewer's suggestion and test Line on graph matching tasks, and it produces inferior results on the 4 smaller datasets, we will mention this method and discuss in the experiment section and append the results to the appendix. The poor performance of LINE can be attributed to the fact that it is not permutation equivariant.
>
> ```markdown
> LINE for Graph Matching
> |Pert|  Celegans  |    Arena    |    Douban  |    Cora    |
> |----|------------|-------------|------------|------------|
> | 0  |  1.0+-0.5  |   0.2+-0.1  |  0.0+-0.0  |  0.0+-0.0  |
> |----|------------|-------------|------------|------------|
> | 1% |  1.0+-0.4  |   0.1+-0.1  |  0.0+-0.0  |  0.1+-0.0  |
> |----|------------|-------------|------------|------------|
> | 5% |  0.9+-0.3  |   0.2+-0.2  |  0.0+-0.0  |  0.1+-0.0  |
> |----|------------|-------------|------------|------------|
> ```

---

> ### Author Response · Authors · 2023-11-17
>
> > The larger datasets you transfer to are known to be structurally similar to the smaller datasets you trained on, e.g., they are also homophilic. It might be more meaningful to consider to what degree your pre-trained models transfer to datasets with drastically different characteristica, such as for example heterophilic datasets. Or whether your method is still viable at even larger scale than 18 thousand nodes.
>
> The reviewer is right that the considered datasets are homophilic. It is also true that the current results on the transferability of GNNs (Ruiz et al., 2020), suggest that we can train with small graphs and efficiently execute with much larger graphs when the substructures (motifs) that appear in the tested graphs, were also partially observed during training. However, graph matching benchmarks do not include heterophilic datasets, as it is unclear whether matching heterophilic graphs is of practical interest. Furthermore, there is nothing that prohibits our approach to perform well on graphs with more than 18k nodes. As mentioned in a previous comment our work performs the largest scale experiments in the context of graph matching, to the best of our knowledge.
>
> > The font in Figure 1 is too small to be comfortably read on a print-out.
>
> We appreciate the feedback and will modify the figure to make the fonts easier to read.
>
> > It is unclear to me what you mean by a GNNs output being unique. It would be good if you could clarify your use of the word unique in the paper. GNNs can produce equal representations of different graphs (see for example [3]) and consequently, one could reasonably claim GNN embeddings to not be unique in the sense that GNN embeddings do not uniquely identify graphs they correspond to.
>
> This is an important point which we clarify in the revised manuscript. By unique we mean that the embeddings are permutation equivariant and that the produce the same embeddings every time one executes the trained GNN. Note that this is not the case in factorization methods, which can produce different embeddings for the same graph when you permute the nodes or even when you don't permute the nodes. To avoid any confusion, we will replace unique with permutation equivariant in the revised manuscript and add this discussion.
>
>
> > There is minor errors in the bold font setting in Tables 3 and 4, e.g., "90.0\pm0.4" in row 1 column 3 of Table 3 and "90.1\pm0.4" in row 1 column 3 Table 4.
>
> Thanks for pointing that out. We will fix them in the revised paper.
>
> > I could not see where you describe or reference the GNN_c, which seems to be one of the backbones of your T-GAE. Could you please add further detail on this model or point me to the part of your paper where this is described?
>
> Thank you for pointing out this possible confusion. GNN_c refers to the model decribed by equation (11) in Appendix B. Please refer to Appendix B for furhter details.

---

> ### Author Response · Authors · 2023-11-22
> **We would like to hear your thoughts on the revised paper**
>
> Dear reviewer,
>
> Thank you for appreciating our work and your constructive feedback. Your comments were to the point and we used them to improve our paper. We would appreciate it if you could find some time to share your thoughts with us.

---

> > ### Comment · Reviewer_vYiA · 2023-11-22
> >
> > Thank you very much for your careful and convincing answers. I struggled to assess your responses without an updated manuscript. However, now, based on your revised manuscript, I see that all my comments have been convincingly addressed and implemented in your revised paper. Consequently, I raise my score.

---

### Official Review · Reviewer_qHgX · 2023-11-03

**Soundness:** 3 good
**Presentation:** 2 fair
**Contribution:** 2 fair
**Rating:** 5
**Confidence:** 4

**Summary:**

Network alignment is the task of establishing one-to-one correspondences between the nodes of different graphs. In this work, the authors propose a graph autoencoder architecture designed to extract node embeddings that are tailored to the alignment task. They prove that the generated embeddings are associated with the eigenvalues and eigenvectors of the graphs and can achieve more accurate alignment compared to classical spectral methods.

**Strengths:**

1. The paper is easy to follow, and the claims made by the authors are very clear.
2. Theorem 3.2 effectively supports their proposed approach, comparing the performance of a GNN to that of the spectral approach.

**Weaknesses:**

1. There is little explanation on their actual model, i.e., Figure 1 and 2. The authors should give the full information of their approach in the main paper, not in the appendix (maybe E.4).
2. In Table 3, the performance improvement over WAlign or FINAL seems negligible. Since the “perturbed” datasets are created synthetically by the authors, there should be more realistic benchmarks that can exhibit the superiority of the approach.
3. Although the authors include 8 graph datasets in the experiments, there is only one setting of experiments; they used specific 4 datasets in the training, and evaluated on all the datasets. Why do the authors pick this setting? Can they perform experiments with more diversity?
4. (Related to Weakness 2) In Section 5.2.2, the authors create the datasets using the same perturbation approach used for their proposed method. The performance improvement is not surprising because of that reason. It might be better to find a real perturbation scenario and see if their proposed way of perturbation can deal with realistic problems.
5. There are too much white space and redundant writing in the paper. For example, Remark 4.1 is unnecessary since the authors repeatedly say about that throughout the paper. Equations (8), (9) and (10) can be also presented more concisely. The authors can bring more important information into the paper if they use the space more efficiently.

**Questions:**

1. The authors claim that “The implications of this framework are of paramount importance since they enable node embedding and graph matching for large-scale graphs, where training is computationally prohibitive.” How does Equation (9) lead to scalability?

---

> ### Author Response · Authors · 2023-11-17
>
> > There is little explanation on their actual model, i.e., Figure 1 and 2. The authors should give the full information of their approach in the main paper, not in the appendix (maybe E.4).
>
> Thank you for the suggestion, we have added details of the proposed approach in the Experiments Section of the revised manuscript.
>
> > In Table 3, the performance improvement over WAlign or FINAL seems negligible. Since the “perturbed” datasets are created synthetically by the authors, there should be more realistic benchmarks that can exhibit the superiority of the approach.
>
> We appreciate the suggestions on our experiments. However, we respectfully disagree that our improvement is negligible. In Table 3, we tested our model and the baseline methods on matching graphs with different levels of perturbation. In the case of no perturbations, most of the baselines work well. With perturbation, our method markedly outperforms FINAL and WAlign. For example, with 1% perturbation, T-GAE outperforms FINAL by 73% at most, and WAlign by 11%. With 5% perturbation, the largest accuracy difference between our method and FINAL, WAlign is 51% and 36% respectively. All the datasets used in Table 3 are real-world networks, and it is true that the perturbations are generated in a synthetic manner. In addition to Table 3, we included, in the original submission, experiments with different real-world networks with partially aligned nodes (Section 5.3, Fig. 3). Note that all the datasets we used are benchmarks for network alignment/graph matching.
>
> > Although the authors include 8 graph datasets in the experiments, there is only one setting of experiments; they used specific 4 datasets in the training, and evaluated on all the datasets. Why do the authors pick this setting? Can they perform experiments with more diversity?
>
>
> We presented 5 settings of experiments to test the effectiveness of the proposed T-GAE: (1) Train on the 4 smaller datasets without perturbation, we train the GNN encoder on the original adjacency matrices of the family of smaller graphs, the results are presented in Table 3 of the manuscript. (2) Train on the 4 smaller datasets with perturbation, we train the GNN on the perturbed adjacencies of the family of training graphs, and the results are summarized in Table 4. (3) Train the GNN on a specific graph, we train the GNN on the original adjacency matrix, and test matching the adjacency with its perturbed versions. (4) Train the GNN on a specific perturbed graph, and test on other randomly generated perturbations. Results for (3) and (4) are presented in Table 2 in the Appendix. We also repeated these 4 settings for the degree perturbation model, where we add or remove edges according to the degree of its nodes, and we refer the reviewer to Appendix Section H for the results of the degree perturbation model. We also conducted real world experiments to match sub-graphs of two different networks, and the results are in Figure 3.
>
> We emphasize the transferable training setting in Table 3 because training on original adjacency matrices is the most generalizable setting and the empirical results show great transferability of the proposed method, which leads to efficient matching on large-scale graphs.
>
> > (Related to Weakness 2) In Section 5.2.2, the authors create the datasets using the same perturbation approach used for their proposed method. The performance improvement is not surprising because of that reason. It might be better to find a real perturbation scenario and see if their proposed way of perturbation can deal with realistic problems.
>
> We agree with the reviewer that in Section 5.2.2 we use the same perturbation model in training and testing. However, the graphs used in training are different than the graphs used for testing as all the perturbed samples are randomly generated. Besides, Section 5.2.2 is just one part of our experiment. In Table 3, the GNN is not trained on the perturbed graphs, but on original adjacency and the proposed method still achieves the best accuracy among the competing baselines.
>
> We agree with the comment that we "should include real perturbation scenario" and that is why we have real-world sub-graph matching experiments presented in Figure 5. It is the most realistic scenario, in which we match 2 different real graphs with partially aligned nodes.

---

> ### Author Response · Authors · 2023-11-17
>
> > Remark 4.1 is unnecessary since the authors repeatedly say about that throughout the paper. Equations (8), (9) and (10) can be also presented more concisely. The authors can bring more important information into the paper if they use the space more efficiently.
>
> Thank you for the suggestion. We agree with the reviewer that we should modify this section to present our method more concisely, and we will modify this in the revised paper. However, we believe that equations (8), (9), and (10) are important. In equation (8) we present the classical graph autoencoders, Equation (9) is the proposed generalized graph autoencoder which is trained on a family of graphs, and Equation (10) presents the proposed T-GAE that is trained on multiple graphs and their perturbed versions from the data augmentation process. Regarding 4.1 please refer to the next response about its connection to transferability.
>
> > The authors claim that “The implications of this framework are of paramount importance since they enable node embedding and graph matching for large-scale graphs, where training is computationally prohibitive.” How does Equation (9) lead to scalability?
>
> Learning efficient node representations for large-scale graphs is a challenging task. Factorization and auto-encoder methods produce powerful embeddings, but the quadratic complexity of the reconstruction (decoder) limits their applicability to small/regularly-sized graphs. To overcome this limitation our idea is to leverage the transferability properties of a GNN (proposed encoder), while maintaining the representation power of auto-encoders. The transferability analysis for GNNs (Ruiz et al., 2020), suggests that we can train with small graphs and efficiently execute with much larger graphs when the substructures (motifs) that appear in the tested graphs, were also partially observed during training. Equation (9) and (10) describe our proposed generalized graph auto-encoder, which is trained with multiple graphs. As a result, a variety of motifs are observed during training, which cannot be observed with a classical graph autoencoder, and the proposed GNN encoder can be transferred to large-scale graphs. As a result, Equation (9) and (10) are the key to the scalability of our approach. This discussion is included in the revised manuscript. Also, this discussion is included in Remark 4.1

---

> ### Author Response · Authors · 2023-11-22
> **We would appreciate your feedback**
>
> Dear reviewer,
>
> Thank you for your super useful comments. Your feedback helped us improve the quality of our paper significantly. We believe we have addressed all of your concerns and would like to hear your thoughts on our response and revised paper.

---

> > ### Comment · Reviewer_qHgX · 2023-11-22
> > **Reply to Authors**
> >
> > Thank you for the detailed response. However, I don't think my concerns are addressed well.
> >
> > My original concerns have two aspects. First, the accuracy improvement over baselines is negligible in Table 3 when there is no perturbation. This implies that the proposed approach is meaningful only in the presence of perturbation. Second, the perturbed datasets are created synthetically, even though they are based on real-world graphs. This leads me to consider the possibility that the proposed approach may be tailored to a specific perturbation scenario that may not be prevalent in the real world.
> >
> > My questions are: (a) Are there any datasets that contain real-world perturbations or noises? (b) Is the authors' approach in Section 5.3 the only way to create and inject anomalies? (c) If not, can we imagine other types of perturbations and demonstrate that the proposed approach can perform well even in such cases, when the nature of perturbation differs from what was assumed in the design of the proposed approach?

---

> > > ### Author Response · Authors · 2023-11-23
> > > **Feedback for the additional clarification**
> > >
> > > We thank the reviewer for their continuous engagement in the discussion which has been fruitful for us. Regarding the clarified concerns and questions, we have given additional clarification and explanation of our method and experiments. We would appreciate any feedback regarding our new comments and the revised manuscript which we have uploaded for the reviewers to check out.

---

> ### Author Response · Authors · 2023-11-22
>
> Thank you for taking the time to respond and engaging in this discussion. We agree with the reviewer that graph-matching algorithms should be tested in real-world perturbation scenarios, and this is why we included them in Section 5.3 of the original manuscript (5.4 in the revised). The results indicate that the proposed T-GAE works remarkably well in real-world graph matching. Please find our detailed responses to your comments below.
>
> >The perturbed datasets are created synthetically, even though they are based on real-world graphs. This leads me to consider the possibility that the proposed approach may be tailored to a specific perturbation scenario that may not be prevalent in the real world.
>
> The proposed approach **is not** tailored to a specific perturbation model. Note that in graph matching perturbations correspond to adding or removing edges. There is nothing else one can do. We consider two such models: one in the main paper, and a second in Appendix G. The results of both Tables 3 and 7, correspond to **training with zero perturbation, but testing on perturbed graphs that follow different perturbation models**.
>
> As we discussed in our initial response, **we also have the experiments of section 5.4 that consider matching 2 real graphs (no synthetic perturbation involved)**. The proposed T-GAE works remarkably well in this task.
>
> >The accuracy improvement over baselines is negligible in Table 3 when there is no perturbation. This implies that the proposed approach is meaningful only in the presence of perturbation.
>
> It is not that the proposed approach is meaningful only in the presence of perturbation, **the task of network alignment (graph matching) is only meaningful in the presence of perturbation (when the considered graphs are not the same)**. When there is no perturbation (the considered graphs are the same) the problem is called Graph Isomorphism and is not proven to be P or NP, and is not the focus of this paper. As a result, the zero perturbation scenario is only discussed as a reference point and we do not assess the performance of any network alignment algorithm based on this.
>
> We also give short answers to the questions raised by the reviewer for clarity:
>
> > (a) Are there any datasets that contain real-world perturbations or noises?
>
> Yes, the datasets of Section 5.4 ACM-DBLP and Douban Online-Offline. Please see our previous response.
>
>  > (b) Is the authors' approach in Section 5.3 the only way to create and inject anomalies?
>
>  No, please refer to our previous responses and section G in the Appendix.
>
>  > (c) If not, can we imagine other types of perturbations and demonstrate that the proposed approach can perform well even in such cases, when the nature of perturbation differs from what was assumed in the design of the proposed approach?
>
>  Yes. Please refer to Tables 3, 7, and Fig. 3. In these experiments there is no perturbation assumed in the design of the approach. In fact, Fig. 3 presents results with purely real graphs.
>
>
> Again, we thank the reviewer for their efforts in clarifying their questions, so that we can better address them, and take the valuable suggestions to improve the presentation of our paper. We have also uploaded the revised manuscript for the reviewer to check out.

---

### Meta-Review · Area_Chair_ggyJ · 2023-12-12

**Metareview:**

The reviewers very much appreciate the effort made by the authors to address their concerns, especially in light of the extensive experiments they included during the discussion phase.

Nevertheless, the paper does have some room for improvement. Reviewers are indeed convinced that the paper either outperforms or is similar to SotA, and that SotA may not scale as well or have other deficiencies (e.g., not handle isolated nodes). Another way of describing what seems to be happening: the proposed algorithm outperforms SotA on specific datasets and remains competitive on other datasets. Reviewers believe that this can be further explored: if results depend on the perturbation model, what is their robustness in a real-life scenario? Is it really the noise or other topological properties of the graph that contribute to discrepancies in performance?

Another issue that remains is the performance under no perturbations: the statement "the task of network alignment (graph matching) is only meaningful in the presence of perturbation (when the considered graphs are not the same." is concerning: graph isomorphism is challenging and has motivated a lot of research in network alignment!  On another front, however, the no perturbation scenario is "easy"; one would expect an algorithm to do well here, and then be robust to perturbations. Another way of phrasing this: an algorithm that is not competitive under no perturbation is not competitive under little perturbation, and it is not clear where the line is to be drawn.

**Justification For Why Not Higher Score:**

Concerns by negative reviewers seem valid to me.

**Justification For Why Not Lower Score:**

N/A

---

### Decision · Program_Chairs · 2024-01-16

Reject